# Intermediates of forming transition metal dichalcogenide heterostructures revealed by machine learning simulations

Luneng Zhao [1,2], Hongsheng Liu [1], Yuan Chang[1], Xiaoran Shi[1], Jijun Zhao [3], Feng Ding[2] & Junfeng Gao [1,2] ✉

Two-dimensional (2D) transition metal dichalcogenide (TMD) van der Waals heterostructures (vdWHs) hold promise for high-performance electronics, but their large-scale synthesis remains limited by size constraints and alloying contaminations. Recently, a two-step vapor deposition method was reported for growing wafer-size TMD vdWHs with minimal impurities. In this study, we develop a machine learning potential (MLP) that captures the atomic-scale dynamic growth process of bilayer $MoS_2/WS_2$ vdWHs under feasible growth conditions. Our simulations uncover a crucial metastable SMMS (M = Mo or W) intermediate structure that facilitates metal atom swap and alloying. Eliminating the alloying contamination requires preventing the embedding of bare metal atoms. The results also show that the SMMS structure exhibits favorable electronic properties and emerges as a low Schottky barrier contact electrode for $MoS_2$ field-effect transistors (FETs).

2D TMDs have been extensively studied due to their semiconducting band gaps and high carrier mobility[1–4], strong nonlinear optical response[5,6], and ease of layer stacking assembly. The integration of vdWHs by pristine TMD layers can further enhance their properties, enabling various potential applications in microelectronics, optoelectronics, and nonlinear optics[7–14].

Despite their promise, the controlled growth of TMD vdWHs faces numerous challenges. While commonly used mechanical assembly methods can achieve high-quality TMD vdWHs with atomically sharp interfaces[15,16], they often struggle to achieve wafer-size dimensions and can be prohibitively expensive. In contrast, chemical vapor deposition (CVD) has achieved significant success in growing wafer-sized monolayer TMD[12,17,18]. However, growing TMD vdWHs still faces limitations in size constraints and a strong tendency toward alloy formation[19,20]. Among these scalable methods, metal-organic chemical vapor deposition (MOCVD) has emerged as a promising alternative for growing large-size TMD heterojunctions with improved interfaces[20–22].

Recently, it was reported that a two-step vapor deposition process with a high-to-low temperature strategy was used to synthesize wafer-sized TMD vdWHs[23]. In this method, a monolayer $WS_2$ film was first achieved by depositing a W film on a sapphire substrate and sulfurizing it at 900 °C, the highest temperature among the four stacked materials. Next, a Mo film was deposited on the $WS_2$ monolayer via magnetron sputtering and sulfurized at 800 °C to form $MoS_2$. Subsequently, a Nb film was deposited on the $WS_2/MoS_2$ film via magnetron sputtering and selenized at 700 °C to form $NbSe_2$. Finally, $PtTe_2$ was grown on the $WS_2/MoS_2/NbSe_2$ vdWHs at 350 °C, the lowest temperature in the sequence. The final structure was a wafer-sized vdWHs several centimetres in size, consisting of four layers: $WS_2/MoS_2/NbSe_2/PtTe_2$. The authors further proposed that, during the metal deposition process, the metal atom films coat the TMD surface[23].

Compared with experimental trials that involve many parameters and various growth conditions, accurate atomic simulations can certainly provide an in-depth understanding of the growth of TMD vdWHs. The growth of TMD vdWHs is always accompanied by rapid

[1]Key Laboratory of Materials Modification by Laser, Ion and Electron Beams (Dalian University of Technology), Ministry of Education, School of Physics, Dalian, China. [2]Suzhou Laboratory, Suzhou, China. [3]Guangdong Basic Research Center of Excellence for Structure and Fundamental Interactions of Matter, Guangdong Provincial Key Laboratory of Quantum Engineering and Quantum Materials, School of Physics, South China Normal University, Guangzhou, China. ✉e-mail: gaojf@dlut.edu.cn

formation and breaking, involving various intermediate motifs with complex chemical bonds, such as metallic bonds, covalent bonds, both covalent and ionic characters of Mo-S bonds, and layered vdW forces. Density functional theory (DFT) is well suitable for describing the complex chemical bonding, but is limited to afford large-scale simulations. In contrast, classical molecular dynamics (MD) can simulate large systems but lacks the capability to capture complex chemical bond recombination. Although various machine learning potentials (MLPs) have been developed and reported to simulate large systems with accuracy comparable to DFT[24–29], only a few MLPs are capable of describing intricate growth processes of TMD vdWHs.

In this study, we developed an MLP trained on extensive DFT data and utilizing a revised equivariant graph neural network implemented in the NequIP package[24]. The MLP was then implemented into MD (MLP-MD), enabling both large-scale and accurate simulation of the $MoS_2/WS_2$ vdWHs growth process. Our results indicate that a bare metal atomic layer is unstable on TMD layers; it spontaneously sinks into the S layer, forming a crucial intermediate structure (SMMS) with high stability. This behavior facilitates the exchange between Mo and W atoms, thereby revealing the atomic-scale mechanism of TMD alloys formation. To grow pristine TMD vdWHs, it is essential to suppress the SMMS formation, which can be achieved by preventing bare metal atom adsorption on pre-existing TMD layers. On the other hand, the SMMS structure serves as an ideal metallic electrode with a low Schottky barrier height, enabling $MoS_2$ integrated circuits through the planar deposition of metal atoms on uncovered $MoS_2$.

## Results

### MLP Development and DFT Validation

To enhance the complexity and versatility of our MLP model, we designed a diverse dataset encompassing a wide range of Mo, W, S, and mixed structures. We also included Se atomic structures to broaden the scope of potential research applications. Through an iterative learning process, we carefully selected and balanced the weights of different structural configurations. The initial training set was derived from on-the-fly machine learning MD simulations[30] and was then continuously enriched with new structures throughout successive iterations. The final dataset included various TMD layers, $MoS_2/WS_2$ vdWHs, $MoS_2$ or $WS_2$ with homogeneous and heterogeneous metal clusters, S clusters, diverse MoWS alloy configurations, and growth intermediate structures [Fig. 1]. This comprehensive approach resulted in a dataset of approximately 26,000 DFT-calculated entries. Detailed procedures for both the initial training and subsequent iterative refinement are provided in the Supplementary Information. Figure S1a demonstrates a high consistency between the MLP and DFT energies, achieving a root mean square error (RMSE) of 10.6 meV/atom. Figure S1b further illustrates a high consistency between atomic forces predicted by the MLP and those from DFT calculations, with an RMSE of 151 meV/Å. To comprehensively assess the model's reliability across the energy landscape, we analyzed the prediction errors as a function of structure formation energy [see Fig. S1c]. This analysis confirms that the MLP maintains high accuracy not only for stable low-energy configurations but also for high-energy states relevant to growth simulations. Overall, despite the structural complexity and complex bond changes in the growth process potentially introducing some noise, these errors are significantly smaller than the thermal energy ($kT$) of ~90 meV at typical $MoS_2$ growth temperatures ( ~ 800 °C). Therefore, our MLP model maintains sufficient accuracy for describing TMD growth processes. These error values are comparable to those reported in recently published MLPs for growth simulations[31,32]. A systematic comparison of energies of key structures obtained from MLP and DFT further confirms the reliability of our model shown in Table S1. It can be seen that a high consistency between MLP and DFT in the relative stabilities of key configurations, such as SMMS and $MS_2$. Specifically, for the SMMS-like configuration generated by direct MD deposition and the constructed and optimized SMMS structure, the energy difference is 0.080 and 0.068 eV, obtained from DFT calculations and MLP predictions, respectively. Similarly, for the SMoWS system, the

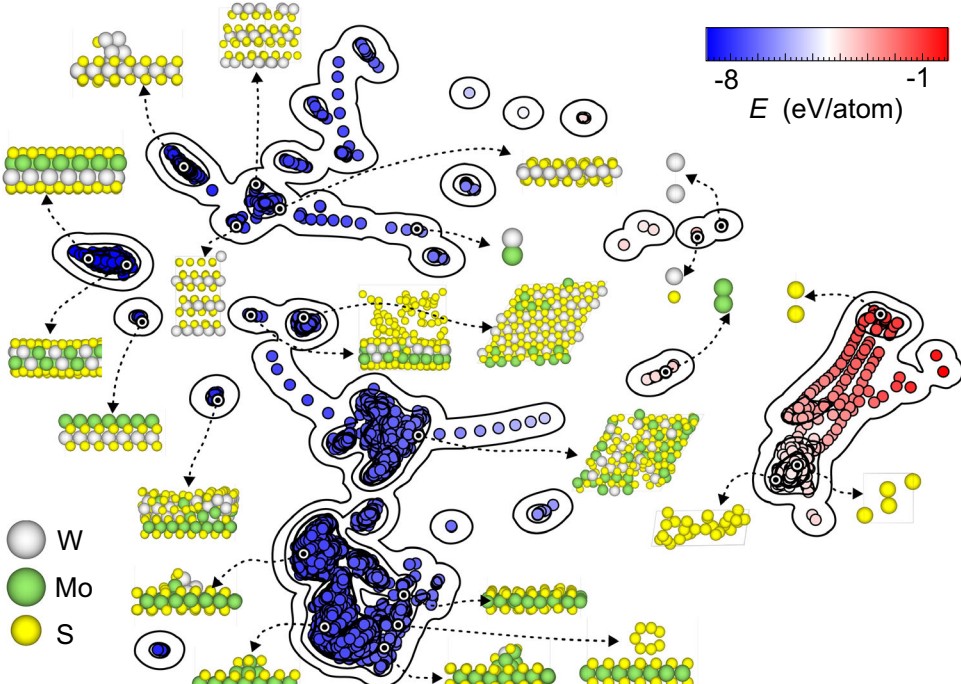

**Fig. 1 | Structural diversity of the dataset.** The key structures within the dataset are visualized using principal component analysis (PCA) to illustrate the diversity of TMDs, $MoS_2/WS_2$ vdWHs, and various complex structures that emerge during growth processes. The density of dataset in specific areas is represented by contour lines. Source data are provided as a Source Data file.

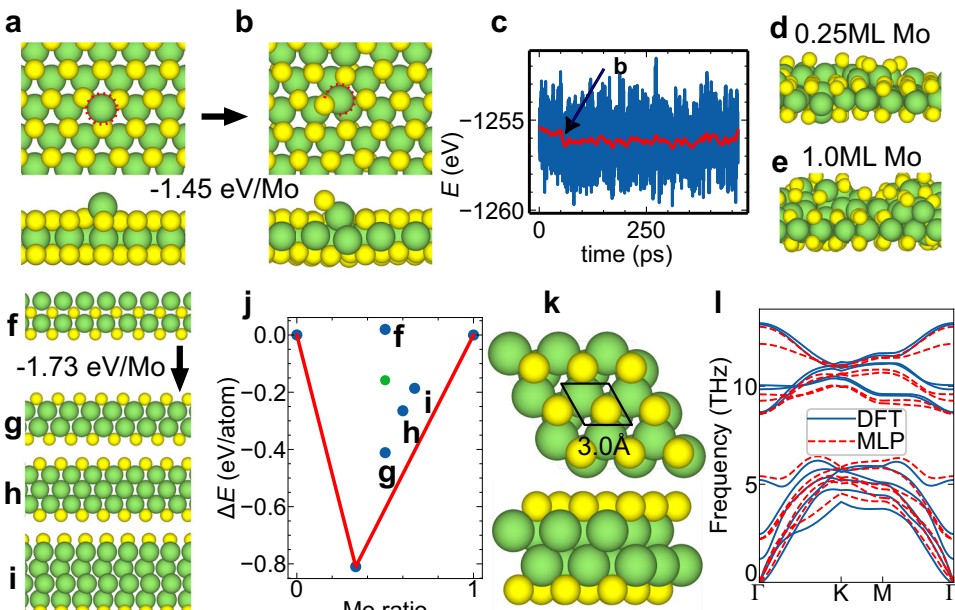

**Fig. 2 | Deposition dynamics and stability of Mo atoms on MoS₂. a** The structure of Mo atoms deposited and **b** embedded into the MoS₂ layer, **c** energy evolution during the MLP-MD simulation at 1100 K (blue line represents raw energy, while red line shows the low-pass-filtered result). **d, e** Snapshots of 0.25 monolayer (ML) **d** and 1.0 ML **e** Mo atom deposition simulated at 900 K on the existing MoS₂ layer. **f–i** Four possible structures: **f** MoSMoS, **g** SMoMoS, **h** SMo₃S, and **i** SMo₄S with more Mo atoms embedded. **j** Formation energy convex hull of the considered MoS structures, with green dots corresponding to energies of structures found in the 2D Materials Database (MatHub-2d). **k** The unit cell structure of the SMoMoS and **l** its phonon dispersion relation. In (**a, b, d, e, f–i, k**), the green and yellow spheres represent Mo and S atoms, respectively. Source data for (**c, j, l**) are provided as a Source Data file.

corresponding energy difference is 0.090 eV from DFT simulation and 0.077 eV from MLP prediction. These results demonstrate that our MLP can compute the energy differences between different structures and reliably describe the relative stabilities of materials.

To further validate the capability of our MLP to simulate the growth of TMD materials using MLP-MD, we randomly mixed Mo, W, and S atoms in a 1:1:4 ratio, then annealed the mixture at temperatures ranging from 1500 K to 900 K for 2 ns. This process resulted in the formation of ordered TMD layers, including the 1H and 1 T phases, demonstrating the reliability of our MLP in simulating the complex layered growth behavior of TMDs [Fig. S2]. An animation of the annealing process is available in the Supplementary Information as Supplementary Movie 1. These results underscore the effectiveness of our iterative learning approach in developing a robust and accurate MLP for simulating the growth of TMD heterostructures.

### Deposition Dynamics of Mo Atoms on MoS₂

Before investigating TMD vdWHs, the developed MLP-MD model successfully simulated the growth of bilayer MoS₂ through a two-step vapor deposition method[23]. First, the adsorbed single Mo atom was unstable on the MoS₂ layer [Fig. 2a] and, at the typical growth temperature of 1100 K, quickly sank beneath the S atom layer within tens of picoseconds [Fig. 2b]. DFT calculations confirm that this embedding process releases 1.45 eV of energy. Figure 2c shows a significant drop in energy at around 50 ps, indicating that the Mo atom has been embedded into the MoS₂ layer. Previous studies have shown that single or paired metal atoms suspended on bare surfaces exhibited significant catalytic enhancement[33–36]. This indicates that the configuration of metal atoms needs to be carefully considered in single-atom catalysis.

Subsequently, individual Mo atoms were simulated to be continuously sputtered onto the existing MoS₂ layer with a kinetic energy of 0.12 eV. To simulate the stability during the metal atom deposition and pre-heating stage, simulations were performed at 900 K. The deposition process of Mo atoms that resulted in 0.25 monolayers

(MLs) [Fig. 2d] and 1.0 ML [Fig. 2e] demonstrated the behavior of Mo atoms on the MoS₂ layer [see Supplementary Movie 2]. Figure S3 depicts the evolution of Mo-Mo bonds during continuous deposition, indirectly reflecting the embedding process. We observed that the number of Mo-Mo bonds increased gradually with continued deposition and reached saturation once deposition stopped. As shown in Fig. S4, when Mo atoms were deposited at higher kinetic energies, such as 1 and 5 eV, the Mo atoms were still embedded in MoS₂. This indicates that the embedding behavior of Mo atoms is not sensitive to kinetic energy. Throughout the MLP-MD simulation, no Mo atoms formed the MoSMoS structure [Fig. 2f]. Instead, all deposited Mo atoms spontaneously sank into the MoS₂ layer. These findings suggest that under practical deposition conditions, Mo atoms preferentially form an SMoMoS structure [Fig. 2g]. This SMoMoS structure was then constructed and optimized using DFT, revealing that it exhibits an energy 0.080 eV/atom lower than that of the directly formed embedded structure from MD simulations [Fig. 2e].

We conducted a detailed comparison of these two structures in terms of energy and dynamic stability. DFT calculations reveal that the energy decreases significantly by 1.73 eV per Mo atom when transitioning from MoSMoS to SMoMoS, indicating a strong thermodynamic driving force towards the formation of the SMoMoS structure. In addition to the SMoMoS structure formed by depositing 1 ML of Mo atoms, we constructed and optimized structures, such as SMo₃S [Fig. 2h] and SMo₄S [Fig. 2i], which may form with an increased number of Mo atoms. The convex hull of MoS compounds was plotted by varying the Mo ratio, referencing the bulk phases of elements Mo and S, as shown in Fig. 2j. As shown in the convex hull, MoS₂ has the lowest energy, while SMoMoS is situated 196 meV/atom above the convex hull. We also found structures that match this elemental ratio from other 2D materials databases (MatHub-2d)[37] [Fig. S5], with their formation energies indicated by green dots in the figure, showing that they are 253 meV/atom higher than the SMoMoS structure. SMo₃S and SMo₄S are 221 meV/atom and 218 meV/atom above the convex hull, respectively, indicating that when approximately 1 ML of Mo atoms is

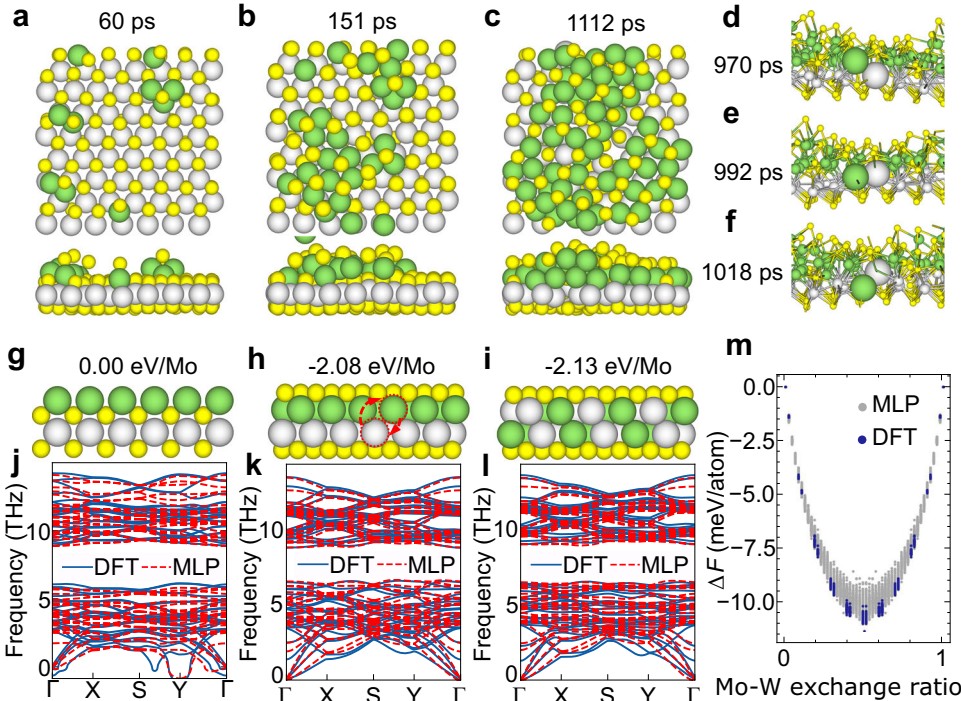

**Fig. 3 | Heterogeneous deposition dynamics and alloying of Mo atoms on WS₂.** **a–c** Snapshots of the growth structure of MoS₂/WS₂ vdWHs during the two-step vapor-deposition process by MLP-MD simulation (Mo atoms are deposited on WS₂) (900 K). **d–f** Observed exchange phenomena of Mo and W atoms during the MLP-MD simulation. **g** Schematic of the Mo layer on the WS₂ surface and **j** related phonon dispersion. **h** Schematic of the unalloyed SMoWS intermediate structure and **k** related phonon dispersion. **i** Schematic of the alloyed SMMS structure and **l** related phonon dispersion. **m** Relative free energy (ΔF) of different Mo-W exchange ratios of the SMMS structure at 300 K. In (**a–i**), the green, gray-white, and yellow spheres represent Mo, W, and S atoms, respectively. Source data for (**j–m**) are provided as a Source Data file.

deposited on the MoS₂ surface, SMoMoS is the most likely intermediate. The phonon dispersion of the SMoMoS structure showed no imaginary frequencies [Fig. 2k and Fig. 2l], further confirming that MoSMoS would spontaneously transform into SMoMoS. It is worth noting that the phonon dispersion simulated by the MLP showed good agreement with the DFT results, validating the accuracy of our MLP in describing structural stability. Long-term MLP-MD simulations further validated the stability of SMoMoS, as shown in Fig. S6.

Furthermore, we used DFT simulations to validate the key processes described above. Figure S7a illustrates the process of a single Mo atom embedding into the MoS₂ layer. The embedding time for vacuum-deposited Mo atoms is approximately 700 fs, contrasting with the ~50 ps required for surface-adsorbed atoms, as indicated by the energy evolution in Fig. S7a. Figure S7b expands our investigation of the behavior of the Mo atomic layer on MoS₂. As the Mo atoms were gradually embedded into the MoS₂ structure, they descend accordingly. This energy decrease is significantly lower than that of the initial configuration, further confirming the thermodynamic favourability of Mo atom embedding. Figure S7c focuses on the stability of the formed SMoMoS structure over approximately 10 ps.

### Heterogeneous Deposition Dynamics of Mo Atoms on WS₂

Following the two-step vapor-deposition process[23], we continuously deposited Mo atoms onto the WS₂ monolayer in MLP-MD simulations, leading to the formation of MoS₂/WS₂ vdWHs [see Supplementary Movie 3]. Figure 3(a–c) display snapshots from the simulation at 0.06 ns, 0.15 ns, and 1.1 ns, showing that the Mo atoms do not remain on the WS₂ surface but are embedded within the WS₂ monolayer. Figure S8 shows the evolution of Mo/W–Mo/W bonds during continuous Mo deposition, indirectly reflecting the degree of Mo atom insertion. The number of these bonds gradually increases with deposition and saturates once deposition stops. Importantly,

during the simulation, Mo and W atoms in the SMoWS intermediate structure layer are able to exchange [Fig. 3d–f], resulting in alloying. The lattice constants of MoS₂ and WS₂ are nearly identical (approximately 1%[23]), reflecting similar behavior for Mo and W. At 300 K [see Fig. S10], simulation results show that while Mo atoms continue to embed into the WS₂ monolayer to form the SMoWS structure, no Mo-W atomic exchange occurs over a 4.5 ns trajectory, indicating that alloying is kinetically suppressed under low-temperature conditions.

Insight into this sinking and subsequent alloying transformation is obtained by analysing relevant energy and phonon dispersions. A single layer of Mo on WS₂ [Fig. 3g] is energetically unfavorable compared to the embedded configuration. Due to the similar interaction between Mo atoms and WS₂, as previously observed with MoS₂, the constructed SMoWS intermediate structure shown in Fig. 3h emerges as another crucial configuration to consider. Its energy is 0.090 eV/atom lower than that of the directly formed structure from MD simulations [Fig. 3c]. Figure 3j shows the presence of imaginary frequencies in the phonon spectrum, indicating that this structure is unstable. Conversely, the Mo atoms completely sink and embed under the top S layer of the WS₂ monolayer, forming the SMoWS intermediate structure [Fig. 3h], releasing 2.08 eV per Mo atom (based on DFT calculations). In addition to the sinking of Mo atoms, atomic exchange between Mo and W atoms occurs, transforming the SMoWS intermediate structure into an alloyed SMMS structure [Fig. 3i]. The phonon spectra of both SMoWS [Fig. 3k] and SMMS [Fig. 3l] show no imaginary frequencies throughout the Brillouin zone, confirming their dynamical stability. Long-term MLP-MD simulations further validate the stability of the SMMS structure, as shown in Fig. S9a and Fig. S9b.

To determine the thermodynamic driving force for alloying, we calculated the free energy change ΔF for varying distributions of metal

atoms between the upper and lower layers. Here, $\Delta F$ is defined as

$$\Delta F = \frac{1}{N}\left[E - E_{\text{SMoWS}} - T\Delta S\right], \qquad (1)$$

where $E_{\text{SMoWS}}$ denotes the energy of the unalloyed SMMS structure [Fig. 3h], $E$ is the energy of the alloyed configuration, $N$ is the number of atoms, $T$ is the temperature (300 K), and $\Delta S$ represents the mixing entropy for Mo/W alloying. The mixing entropy $S$ is given by

$$S = (N_{\text{Mo}} + N_{\text{W}})k_b\left[-x\ln(x) - (1-x)\ln(1-x)\right], \qquad (2)$$

where $k_b$ is the Boltzmann constant, $x$ is the fraction of Mo atoms in the upper metal layer, and $N_{\text{Mo}}$ and $N_{\text{W}}$ are the numbers of Mo and W atoms, respectively. By randomly exchanging metal atoms between the upper and lower layers in the $(7 \times 7)$ SMMS structure supercell and optimizing the structure, we obtained $\Delta F$ for the SMMS structures with different alloy proportions [Fig. 3m]. We found that when the different metal atoms are uniformly distributed between the upper and lower layers, the free energy is minimized; specifically, the energy of $S(Mo_{0.5}W_{0.5})(W_{0.5}Mo_{0.5})S$ (alloyed SMMS) is approximately 11.3 meV/atom lower than that of the unalloyed SMoWS structure. This energy reduction is primarily attributed to the configurational entropy, contributing 9.0 meV/atom, and the alloy structure also alleviates stress caused by structural asymmetry. Furthermore, the relaxed SMMS structure exhibits a lattice constant approximately 5% smaller than that of monolayer $WS_2$. As a result, the deposition and embedding of sufficient Mo atoms into the $WS_2$ substrate generate accumulated stress, leading to potential cracking of the underlying metal-S layer and subsequently accelerating atomic exchange between Mo and W.

In the above-mentioned study[23], a two-step vapor deposition process was reported to synthesize wafer-scale TMD vdWHs, where the key intermediate structure during growth is a Mo atomic monolayer on the $WS_2$ surface. However, our MLP-MD simulations reveal that the Mo atomic monolayer on the $WS_2$ surface is highly unstable, it can be easily transformed into the SMoWS structure by thermal annealing. This suggests that the experimentally observed clean interfaces must be governed by kinetics or environmental factors that prevent the formation of this bare metal intermediate.

## Sulfurization of Intermediate Structures

We now pose the question of what happens when S is deposited on the SMoMoS and alloyed SMMS intermediate phases during the second step. For SMoMoS, a sufficient amount of S atoms was further deposited on the top surface of the SMoMoS intermediate phase [Fig. S11a–e and Supplementary Movie 4]. The simulation shows that the SMoMoS intermediate phase was initially very stable. After 531 ps [Fig. S11b], we observed that S atoms penetrated the SMoMoS structure and pulled some Mo atoms to the surface, gradually forming a bilayer of $MoS_2$ [Fig. S11e]. As shown in Fig. S12a, there are very few initial Mo-S bonds because the S atoms deposited on the SMMS surface cannot bond with Mo. As the MD simulation progresses, some Mo atoms are pulled to the surface to bond with the deposited S atoms, leading to an increase in the number of Mo-S bonds. Between 700 and 1000 ps, a large number of Mo atoms are pulled to the surface, resulting in a substantial increase in Mo-S bonds. The number of Mo-S bonds tends to be saturated after about 1.2 ns, when a bilayer $MoS_2$ is fully formed.

Similarly, for the alloyed SMMS intermediate phase, a sufficient amount of S atoms was further deposited on the top surface

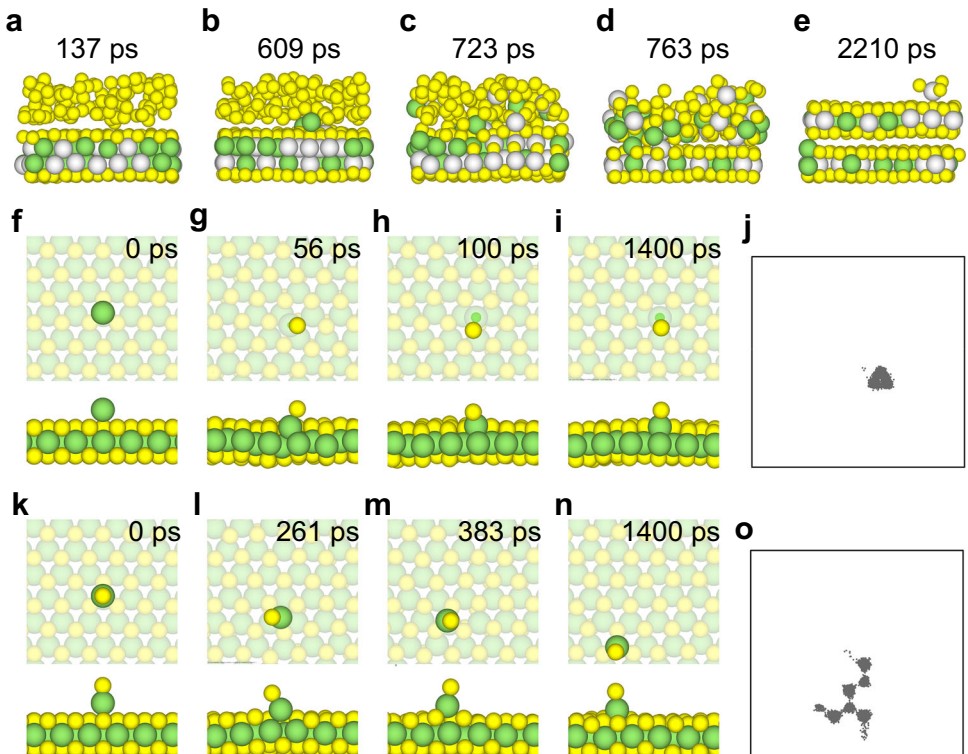

**Fig. 4 | Sulfurization dynamics and surface behavior of Mo-S clusters. a–e** MLP-MD simulations of the growth of alloyed $Mo_xW_{1-x}S_2/Mo_{1-x}W_xS_2$ vdWHs by depositing S atoms on the alloyed SMMS intermediate phase (1100 K). **f–o** MLP-MD simulations of various Mo clusters on $MoS_2$ (1100 K): **f–i** deposition of single Mo atoms; **j** trajectories of Mo atoms in the $xy$ plane; **k–n** deposition of Mo-S clusters and **o** trajectories of Mo atoms in the $xy$ plane. In (**a–i, k–n**), the green, gray-white, and yellow spheres represent Mo, W, and S atoms, respectively. Source data for (**j, o**) are provided as a Source Data file.

[see Supplementary Movie 5]. As shown in Fig. 4a, the alloyed SMMS intermediate phase was also very stable initially. After 608 ps of simulation, S atoms penetrated the SMMS structure and pulled metal atoms from the top surface [Fig. 4b]. However, in the alloyed intermediate phase, S atoms did not selectively extract either Mo or W atoms, instead, they pulled both Mo and W atoms from the upper layer to the surface [Fig. 4c–e]. Therefore, the resulting structure was not a distinct $MoS_2/WS_2$ vdWHs, but rather an alloyed $Mo_xW_{1-x}S_2/Mo_{1-x}W_xS_2$ vdWHs. As shown in Fig. S12b, the evolution of Mo/W-S bonds clearly illustrates this process. According to the experimental results of Zhou et al.[23], the successful synthesis of non-alloyed $MoS_2/WS_2$ heterostructures suggests that the formation of the SMMS intermediate state may have been avoided through process control during the experiment. In the scenario where the intermediate state is the SMoWS structure, we further conducted sulfurization MLP-MD simulations. As shown in Fig. S11f–j, during the simulation, the upper Mo atoms detached from the SMoWS structure. S atoms quickly occupied the original positions of the Mo atoms, effectively preventing the lower W atoms from being extracted. Following the sulfurization of SMoWS, a non-alloyed $MoS_2$ layer formed on the upper layer. This mechanism provides an atomic-level explanation for the experimentally observed non-alloyed growth.

To achieve high-quality, non-alloyed TMD vdWHs, it is essential to prevent Mo atom sinking and the formation of SMMS intermediate phases during growth. We employed MLP-MD to study the behavior of a single Mo atom and various Mo-S clusters on a $MoS_2$ substrate. A bare Mo atom can quickly sink into the $MoS_2$ monolayer, and once submerged, the Mo atom remains firmly embedded with no surface diffusion observed throughout the simulation [Fig. 4f–i]. The trajectory projection in the $xy$ plane [Fig. 4j] further confirms this, showing that the Mo atom is trapped at its initial position throughout the 1.4 ns simulation. In contrast, once a Mo atom bonds with an S atom, the Mo-$S_1$ structure remains on the surface without sinking for the duration of 1.4 ns [Fig. 4k–n]. The structure of Mo-$S_1$ exhibits slightly higher surface diffusion (moving one step every 200 ps), primarily between adjacent Mo top sites [Fig. 4o]. As the amount of S in the structure increases (Mo-$S_2$ and Mo-$S_3$), the surface mobility progressively enhances without embedding [Fig. S13]. This finding indicates that providing an excess of S is critical to preventing the sinking of bare metal atoms and subsequent exchange. To deeply understand the dynamic behavior of Mo atoms and their clusters with S atoms on various TMD substrates, we conducted further MLP-MD simulations to investigate the migration mechanism and stability of Mo clusters on $WS_2$ surfaces. We found that their behavior was similar to that observed on $MoS_2$ substrate (Fig. S14). Figures S15 and S16 illustrate the behavior of single Mo atoms and Mo-S clusters on $MoS_2$ and $WS_2$ surfaces at a lower temperature (900 K). The results indicate that, even at reduced temperatures, single Mo atoms still tend to rapidly embed into the substrates. However, the surface mobility of Mo-S and Mo-$S_2$ clusters is significantly reduced compared to their behavior at 1100 K. In contrast, Mo-$S_3$ clusters maintain relatively high surface stability and mobility, further validating the critical role of sulfur-containing clusters in suppressing Mo atom sinking [Fig. S15, Fig. S16]. Figure S17 further confirms the reliability and accuracy of these MLP-MD results through AIMD simulations at 1100 K for 10 ps. Although the simulation duration was shorter due to computational limitations of AIMD, the clusters displayed similar characteristics as observed in the MLP-MD simulations. Furthermore, these sulfur-rich structures exhibit faster surface diffusion, which is beneficial for the nucleation and aggregation of TMD layers, thereby accelerating defect healing. Experimentally, the sulfur source and molybdenum source ratios used during the MOCVD growth of $MoS_2$ and $WS_2$ are significantly higher than 2:1, such as 70:1[38], 660:1[39], 6400:1[40], and 11111:1[41]. Similarly, regarding the successful synthesis in the above-mentioned study[23], we propose that sulfur-rich conditions likely played a decisive role. Since the Mo film is

deposited after the synthesis of the first $WS_2$ layer, residual sulfur remaining in the growth chamber or on the surface from the preceding step is inevitable. This residual sulfur can react with deposited Mo atoms to form Mo-S clusters during the deposition or the initial heating phase, effectively suppressing the embedding of bare metal atoms and the formation of the SMMS intermediate, as predicted by our simulations. This mechanism explains how alloying is prevented in the experimental two-step process.

By co-depositing Mo and S atoms to form MoS clusters, the homogeneous epitaxy and heteroepitaxy of the second layer of $MoS_2$ on $MoS_2$ and $WS_2$ were simulated at 1100 K, respectively. These simulations are analogous to the MOCVD growth of $MoS_2$ [see Supplementary Movies 6, 7]. As shown in Fig. S18c, the simulation starts by placing a triangular 1H-$MoS_2$ nucleus on the surface of the monolayer 1H-$MoS_2$, representing the nucleation at the onset of growth. The Mo and S atoms required for the growth of one layer of $MoS_2$ were deposited onto the surface within 516 ps, after which the deposition was stopped. Figures S18d–f display the system from 106 ps to 400 ps; the $MoS_2$ layer exhibits characteristics of the 1 T phase. After 2 ns of simulation, the growth of the second layer of $MoS_2$ on the monolayer $MoS_2$ is essentially complete, as shown in Fig. S18g. The 1 T phase present in the early stages diminishes, indicating that $MoS_2$ transitions from the 1 T phase to the more stable 1H phase as growth proceeds. The phase transition is accompanied by the healing of structural defects. The simulation of growing the second layer of $MoS_2$ on a monolayer $WS_2$ proceeds similarly, as shown in Fig. S18h–l, producing a growth pattern characteristic of the 1 T phase [Fig. S18j–k]. As the simulation progresses further, the 1H phase of $MoS_2$ eventually forms on the $WS_2$ substrate, as shown in Fig. S18l.

## Electronic Properties and Device Potential of Intermediate Structures

As previously discussed, the SMMS intermediate structure impedes the growth of non-alloyed $MoS_2/WS_2$ vdWHs. Figure 5a and Fig. 5b show the electronic band structures. Both SMoMoS and SMMS exhibit metallic properties.

We further investigated the contact characteristics of these metallic intermediate structures with the semiconductor $MoS_2$. In both cases, the contact between the metallic intermediate structures and the semiconductor $MoS_2$ forms a p-type Schottky contact, as shown in Fig. 5c, d. The Schottky-Mott limit predicts p-type SBHs of 0.55 eV for SMoMoS and 0.69 eV for alloyed SMMS. For conventional metal electrodes (e.g., Ti, Cr, Au, Pd) interfaced with $MoS_2$, the predicted SBH ranges from 0.56 eV to 1.86 eV[42]. However, Fermi-level pinning induces strong n-type behavior at $MoS_2$ interfaces. Experimentally observed p-type SBHs range from 1.56 to 1.75 eV. We constructed SMMS-$MoS_2$ and SMoMoS-$MoS_2$ interfaces. Compared to Schottky-Mott predictions, we observed significantly reduced Fermi-level pinning (Fig. S19). The p-type SBH increased by only approximately 0.1 eV in the same theoretical framework. Moreover, continuous SBH tuning can be achieved by adjusting metal deposition parameters, enabling tailored contact properties for electronic or optoelectronic applications. These results suggest that SMMS and SMoMoS are promising candidates for $MoS_2$ FET electrodes.

A very recent experimental study[43] realized an atomic layer bonding (ALB) contact by establishing a metallic coherent bonding interface between the transition-metal layer of TMDs and metal electrodes. This ALB structure exhibits direct metal-metal bonding, consistent with the observations in the predicted SMMS intermediate. Consistent with our findings that SMMS exhibits metallic character and improved contact properties, the experimental ALB contact demonstrated ultralow contact resistance and high thermomechanical stability. This strongly corroborates our theoretical prediction that establishing coherent metal-metal interactions (as in SMMS) is an

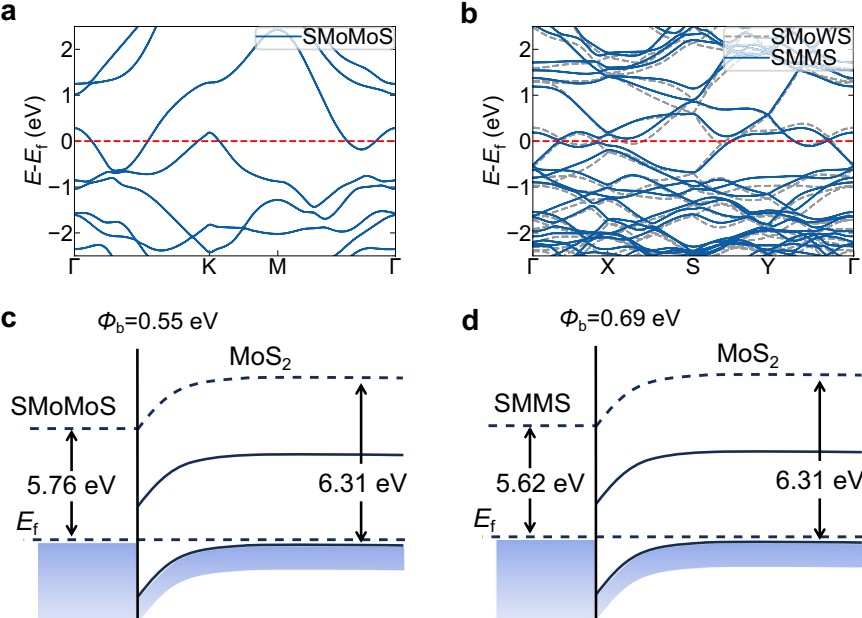

**Fig. 5 | Electronic structures and contact properties of SMoMoS and SMMS electrodes with MoS₂.** **a** Electronic band structure of the SMoMoS structure ($E_f$ is the Fermi level). **b** Electronic band structures of the SMoWS structure and alloyed SMMS structure ($E_f$ is the Fermi level). **c** Schematic of the p-type Schottky barrier (SBH) at the MoS₂-SMoMoS interface, with $\Phi_b = 0.55$ eV. **d** Schematic of the p-type SBH at the MoS₂-SMMS interface, with $\Phi_b = 0.69$ eV. Source data for (**a**, **b**) are provided as a Source Data file.

effective strategy to overcome the limitations of van der Waals contacts and achieve high-performance electronic devices.

## Discussion

In summary, we employed MLP-MD simulations to investigate the growth mechanisms of TMD vdWHs, with a particular focus on the heterostructures of MoS₂ and WS₂. Our study revealed the stability and transformation mechanisms of the intermediate structures SMoMoS and SMMS formed during growth. Through detailed energy and kinetic stability analysis, we observed that these intermediate structures exhibit significant stability under specific conditions and transition to bilayer TMD structures upon further deposition of S atoms. Additionally, we studied the contact characteristics of these metallic intermediate structures with the semiconductor MoS₂ and discovered that they exhibit low p-type SBHs, indicating their potential applications in electronic devices. These findings provide critical theoretical insights into the growth mechanisms of TMD vdWHs and optimization of growth conditions while offering further perspectives for designing 2D materials and devices.

## Methods

### DFT

DFT calculations were performed using the Vienna Ab initio Simulation Package (VASP)[44,45] version 6.3.0. A plane wave basis set was employed, utilizing the projector augmented wave (PAW) method and standard pseudopotentials. The Perdew-Burke-Ernzerhof (PBE)[46] exchange-correlation functional within the Generalized Gradient Approximation (GGA)[47] was employed to compute the electronic structure and energy. The DFT-D3(Becke-Johnson)[48,49] van der Waals correction was chosen to account for dispersion interactions. A plane-wave cutoff energy of 500 eV and no symmetry constraints were applied. To ensure accuracy, the electronic self-consistent loop was converged to a tolerance of $10^{-5}$ eV. Gaussian smearing with a width of 0.05 eV was employed to facilitate the convergence of calculations. Spin-polarized calculations were performed. For all periodic structures, a center-symmetric k-point mesh with a density of 0.3 Å⁻¹ was employed. The

electronic band structure calculations were enhanced through the implementation of the HSE06 hybrid functional for both band structures and electrostatic potentials. A Fermi-Dirac smearing parameter of 0.2 eV was employed in the calculations. MD simulations were conducted using the on-the-fly machine learning potential from the VASP package, extracting frames from DFT calculations as data to accelerate the construction of the initial training set.

### MLP

The MLP was trained using the NequIP[24] framework, which implements E(3)-equivariant graph neural networks. This network provides enhanced stability for MD simulations relative to other networks, enabling extended stable simulations. The model utilized 12 radial basis functions and a maximum angular momentum of 2 for the interatomic edges. The hidden layers included irreducible representations as follows: 128×0e, 64×0o, 128×1o, 64×1e, 32×2o, and 32 × 2e. Element information was embedded as 128-dimensional vectors. Five graph convolution layers were implemented, including self-connections and residual connections. The cutoff distance for constructing the graph was set at 6 Å. The loss function included both energy per atom and atomic force terms, each contributing equally to the total loss. Each loss term used mean squared error. During the initial training, we used a learning rate of 0.005 and a batch size of 5. For the iterative learning process, we employed a lower learning rate of $10^{-4}$ to fine-tune the model. To enhance the model's robustness, we added a constant repulsive term to the output layer:

$$E_{\text{repulsive}} = \left(\frac{r_0}{r}\right)^{12} \frac{r_0}{24} \text{cutoff}(r)$$

This approach enhanced the model's stability, with $r_0 = 1.8$ Å in this study. The iterative learning process allowed us to continuously refine the model by incorporating new structures and configurations, ensuring comprehensive coverage of the complex structural landscape involved in TMD heterostructure growth.

## MD

MD simulations were conducted using the Large-scale Atomic/Molecular Massive Parallel Simulator (LAMMPS)[50], utilizing our developed MLP. We employed the bin algorithm for neighbor list construction with a cutoff radius of 6.0 Å and a skin distance of 2.0 Å. The neighbor list was rebuilt only when at least one atom had moved beyond half of the skin distance threshold. Simulations were conducted within the canonical ensemble (NVT) employing a Nosé-Hoover chain thermostat[51,52], with a temperature damping time set to 0.1 ps. An integration time step of 1.0 fs was employed to ensure performance while maintaining simulation stability. Initial velocities were sampled from a Gaussian distribution. The degrees of freedom contributing to the system temperature were dynamically updated with the deposition of new atoms. The simulations employed an ideal Morse substrate $(V(r) = D_0(1 - e^{-\alpha(r-r_0)}))$ with parameters $D_0 = 0.2$ eV, $\alpha = 1.5$, and $r_0 = 3.5$ Å. This wall mimics the weak van der Waals interaction ($< 0.1$ eV/atom) between Mo/W atoms and oxide substrates, such as sapphire, thus providing only mechanical support while avoiding spurious interfacial chemistry. For Mo atom deposition, we used a $4\sqrt{3} \times 7$ $MoS_2$ or $WS_2$ supercell at 900 K. This temperature was chosen to represent the thermal state during the pre-heating/baking stage and the ramp-up process prior to full sulfurization. The SMoMoS and SMMS sulfurized process simulations were performed using a $4\sqrt{3} \times 7$ supercell at 1100 K, aligning with the experimental sulfurization temperatures (approx. 1073 K). The diffusion behavior of Mo-S clusters on the $MoS_2$ surface was also investigated using a $4\sqrt{3} \times 7$ $MoS_2$ supercell at 1100 K. For the simultaneous Mo and S deposition to grow the second $MoS_2$ layer, we employed a larger $6\sqrt{3} \times 8$ $MoS_2$ or $WS_2$ supercell at 1100 K. The atomic deposition was achieved through the fix deposit command in LAMMPS, which introduced a Mo atom with a downward velocity of 5 Å/ps at random positions within the top region of the simulation box every 6000 steps. The initial kinetic energy of deposited atoms was set to 0.12 eV. To ensure the reproducibility of the non-equilibrium processes, all key qualitative observations (including the sinking of Mo atoms and the sulfurization-induced extraction) were confirmed in three independent MD simulations initialized with different random velocity seeds.

## Reporting summary

Further information on research design is available in the Nature Portfolio Reporting Summary linked to this article.

## Data availability

The training data, trained models, and computation setting files have been uploaded to the Zenodo repository (https://doi.org/10.5281/zenodo.18397127). Source data are provided with this paper.

## Code availability

The training program has been uploaded to a public GitHub repository (https://github.com/1713175349/ocp).

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

## Acknowledgements

The authors acknowledge the financial support provided by the National Key R&D Program of China (2024YFA1409600 to J.G.), R&D project of Joint Funds of Liaoning Province [2023JH2/101800038 to J.G.], the Dalian Science and Technology Innovation Fund (2025JJ12GX012 to J.G.), the National Natural Science Foundation of China (12374253 to J.G., 12374174 to H.L., 12504315 to Y.C.), and the National Foreign Expert Project (D20240213 to J.G., D20240220 to H.L.). Gao thanks Z G Yu (A*STAR) for great helpful discussions and suggestions. The authors also acknowledge Computers supporting from Shanghai Supercomputer Center.

## Author contributions

J.G.: Conceptualization, Supervision, Funding acquisition, Writing – review & editing. L.Z.: Methodology, Investigation, Formal analysis, Writing – original draft. F.D.: Supervision, Writing – review & editing. H.L.: Funding acquisition, Writing – review & editing. Y.C., X.S., and J.Z.: Writing – review & editing.

## Competing interests

The authors declare no competing interests.
