## [Transparent Peer Review file · Nature Communications]

Intermediates of Forming Transition Metal Dichalcogenide Heterostructures Revealed by Machine Learning Simulations

Corresponding Author: Professor Junfeng Gao

Version 0:

Reviewer comments:

Reviewer #1

(Remarks to the Author)

In their manuscript, Zhao and coworkers conduct extensive atomistic simulations to explore the growth process of 2D transition metal dichalcogenide (TMD) van der Waals heterostructures (vdWHs). They employ a machine learning potential (MLP) based on the NequIP architecture, trained on DFT calculations, using an iterative training approach to generate the training and test sets. Non-equilibrium molecular dynamics simulations are utilized to examine the stability and growth of MoS₂/WS₂ heterostructures. The study identifies two distinct growth mechanisms: a sulfur-deficient growth via an SMeMeS (Me=W/Mo) intermediate, and a sulfur-rich growth where the MeS₂ structure remains intact while MeS_x clusters adsorb and agglomerate to form the next MeS₂ layer. The former leads to alloy formation, while the latter prevents metal mixing in the vdWHs. These findings help rationalize a recently published "two-step vapor-deposition process." Additionally, the authors propose that the SMeMeS intermediate could be used to create an atom-scale p-type Schottky contact, potentially significant for technological applications.

The manuscript provides valuable scientific insights into the growth process and should be of significant interest to the 2D-material community. However, there are several shortcomings that require major revision. Firstly, the explanation of the training procedure lacks critical information necessary for reproducibility. Secondly, the MLP validation does not ensure the required accuracy for the results obtained. Thirdly, the non-equilibrium simulations are poorly explained and seem to lack a systematic approach, which raises questions about their robustness. Additionally, the writing is sometimes vague and difficult to follow, this applies to both the language and the organization of the results. I recommend that the authors address these issues to provide a much needed confidence in the presented results (which I believe at the moment to be insufficient) and improve readability.

Major comments:

1. Details of the iterative training procedure for machine learning potential are missing. I couldn't find any information on how they prepared the many different configurations in the training set – and what structures were specifically included. I believe that this also needs motivation since a repository alone leaves too many open questions. The current information on the procedures "on-the-fly MLPMD in VASP" and "Monte Carlo method" are not clear and do not offer information on structural data. Also the data-selection procedure (each iteration with 50% of new data being included) is unclear since "new" data is not defined. Note, that 50% of 100,000 configurations per iteration clearly exceeds a final data set of 26,000 entries
2. The MLP validation consists of fairly meaningless RMSE's and some MD simulations validating(?) the MLP and AIMD validating the stability predictions. These calculations have limited certainty of the accuracy. What I believe is much more important is, e.g. comparing the predicted relative stability of optimized and non-optimized structures (e.g. SMeMeS vs. MeS₂) via the MLP vs. DFT.
3. The non-equilibrium MD simulations are also unclear. How exactly are new species introduced during the MD simulations? Why are they introduced with low velocities. Furthermore, why didn't the authors collect statistics and clearly quantify e.g. mean times of incorporation. The efficient MLP would certainly allow for this. Currently, all results are qualitative observations without any quantitative indication.
4. The discussion on the growth mechanism should be more organized. Each paragraph appears as a loose description of what simulation was performed and what was seen lacking motivation that would help the reader gaining the very important scientific insight.
5. In illustrate the latter point, data from various simulations appears sometimes erratically combined. An example is the discussion of the stability of the bilayer system. If I understood correctly, the structure for MoSMoS, SMoMoS, SMO₃S,

SMo4S (Fig2. (f-i)) are obtained by geometry optimization, not by MD simulation. Did the authors observe the SMOoS (Fig.2 (g)) structure during the MD simulation starting from the MoSMoS structure (Fig.2 (f))? If not, the statement of “all deposited Mo atoms spontaneously sank into the MoS2 layer, tending to form the intriguing SMOoS structure, as shown in Fig. 2(g).” and “In addition to the SMOoS structure formed by depositing 1 ML of Mo atoms” might be misleading, because as seen in Fig. S5 (b) it is less likely that the formation of SMOoS structure from MoSMoS structure happens.

Also, in this discussion, the authors showed a lot of data:

- energy landscape of MoSMoS, SMOoS, SMO3S, SMO4S structures (Fig2. (f-i))
- phonon band structures (Fig2. (l))
- tendency that Mo atoms sink beneath the S layer during the MLP-MD simulation (Fig2. (e))
- tendency that Mo atoms sink beneath the S layer during the AIMD simulation (Fig S5(b))
- stability of SMOoS during the MLP-MD simulation (Fig S3 (a)(d))
- stability of SMOoS during the AIMD simulation (Fig S5 (c))

As such, the data from geometry optimization and MD simulations using DFT and MLP are given, and it is hard to understand how they complement each other in the current version of the manuscript. This should be more clarified for better discussions.

Minor comments:

- Please check the numbering for the figures. For example, Fig. S4 comes earlier than Fig. S3, which is not kind to readers.
- Small comments on the figures:
 - Fig.2 (j) it is stated that “with green dots” but I couldn't find the green point
 - Fig.2 (c) what is the red line and blue line?
 - For the caption for Fig.2, the labels (a) to (e) come after the caption but (f) to (l) come before the caption, which is, I think, very confusing.
 - Fig.5 (b) SMOWS should be SMOoS
- For the phonon band structures, the data from DFT and MLP are plotted together on the same figure, but I suggest to show only DFT results. The authors stated in page 3 “It is worth noting that the phonon dispersion simulated by the MLP accurately reproduced the DFT results, validating the accuracy of our MLP in terms of atomic forces.”, but some degeneracy at the gamma point are not well described by MLP, and it is hard to tell the accuracy of MLP from this figure. For Fig.3 (j-l), the authors don't mention the differences between DFT and MLP, so I suggest removing the phonon band structures by MLP for better clarity.
- Equations in Page 4
 - Please define N and T.
 - E_SMOoS is energy, then why is E formation energy?
 - x should be “one Mo atom” not “one metal atom”?
- Page 5: “Throughout the calculations, general precision was used”, it would be better to specify the precision. “general precision” is vague.
- For the simulation given in video S4, the S layer looks floating above the SMOoS layer. Is it constrained?
- Page 5, what do you mean by “a specific rate of 516 ps per layer”?
- For most of the systems, the system size is not clearly stated but it should be given.
- Fig. S5(c) shows the AIMD simulation of SMOoS structure until 10 ps, but what happens after 10 ps? Did the system retain the SMOoS structure?
- Again, are the structures of Fig.3 (g-i) obtained by geometry optimization and not appeared during the MD simulations?

(Remarks on code availability)

Reviewer #2

(Remarks to the Author)

(Remarks on code availability)

The manuscript by Zhao et al. presents a machine learning study investigating the atomistic details of van der Waals heterostructure (vdWH) formation during chemical synthesis. While the topic is inherently interesting and relevant, the paper does not meet the necessary criteria for publication due to several significant shortcomings.

First, the language throughout the manuscript, particularly in the first two pages, is poorly constructed, making it difficult—at times even impossible—to discern the authors' intended meaning. This lack of clarity severely impacts the readability and comprehension of the work.

Second, the overall structure and logic of the manuscript are unclear. The progression of ideas is disorganized, making it challenging to follow the content from beginning to end. Additionally, several critical details are omitted, which undermines the rigor of the study. For instance:

- (i) The temperatures at which the calculations were conducted are not clearly specified.
- (ii) The substrates used for investigating the formation of vdWHs are not identified.

The authors motivate their theoretical study with experimental results published in Nature (621, 499, 2023). However, they

fail to provide sufficient a meaningful comparison with their theoretical results with the experiment. Such a comparison is essential for evaluating the reliability of the study and its implications for the synthesis of pure vdWHs (without alloying).

In its current state, the manuscript lacks the clarity, structure, and critical details necessary for a proper review. Given these substantial deficiencies, I regret to conclude that the manuscript cannot be considered for publication and must be rejected.

Reviewer #3

(Remarks to the Author)

(Remarks on code availability)

Reviewer #4

(Remarks to the Author)

The authors have written an interesting paper on the formation of a stable intermediate state which they call SMMS that forms when depositing bare Mo atoms on MoS₂ or WS₂. While there isn't much methodological innovation in the paper, at least as currently presented, it is a nice and fairly complex application of ML potentials to an important class of materials that sheds some light on the formation of MoS₂/WS₂ bilayers, namely that Mo must be deposited with sulfur to ensure that Mo doesn't sink below the surface.

I do have some concerns that I think should be addressed if the paper is to be published in Nature Communications.

Major concerns

* The iterative learning approach presented in the paper is unclear. The results section states that the initial training set was generated with on-the-fly learning and then "continuously enriched with new structures throughout the iterations". The methods and SI do not provide much additional information, beyond the number of iterations (17) and the number of "data points" per iteration (100,000). Are these data points entire structures or individual forces and energies? How exactly were these new structures generated? The SI alludes to a Monte Carlo method, but this is vague. In my opinion, the generation of the training set is central to the paper and will be of interest to others in the field. I suggest the authors describe their method in much more detail. As presented currently, it is not possible to reproduce it.

* One of the central claims of the paper, appearing in the last sentence of the abstract, is that the SMMS phase that they identify is an "ideal electrode for MoS₂ FETs". This claim is weakly supported. The two pieces of evidence for the claim are 1) SMMS is a metal, as evidenced by the band structure in Fig. 5a-b, and 2) the p-type Schottky barriers seem to be in a reasonable range. To evaluate the claim that this structure is an "ideal electrode", much more information would need to be presented, including comparison with existing electrodes, assessment of how easy SMMS is to synthesize relative to these other electrodes, etc. The authors also seem to suggest that the difference between 0.55 and 0.69 eV (the Schottky barriers for SMOmOS and SMMS respectively) is significant, but this seems hard to believe without additional context.

Minor concerns

* The authors claim in the SI that the accuracy of their potential is "outstanding", but the RMSEs don't strike me as particularly remarkable (151 meV/Å on forces and 10.6 meV on energies). They also claim that their potential "outperforms several existing machine learning force fields in key aspects", but do not elaborate. For these claims to stand the authors will need to provide direct comparisons with competing potentials. Otherwise, the claims should be removed or significantly weakened.

* It seems the authors do not train on the stress tensor, which is unusual. Why were stress labels excluded from the training set?

* In the results section, the authors claim that "the phonon dispersion of the SMOmOS structure showed no imaginary frequencies, further confirming that MoSMoS would spontaneously transform into SMOmOS." Lack of imaginary frequencies only confirms that SMOmOS is stable, not that MoSMoS would transform into it.

* In the results section, when comparing to AIMD simulations, the authors claim that the time for Mo to embed below the surface is significantly shorter than in MLMD simulations "due to the high chemical reactivity of the Mo atoms from vacuum". I don't understand this explanation. Why isn't this effect captured in the MLMD simulations, which are supposed to reproduce AIMD results?

* The convex hull presented in Fig. 2j is unclear. The caption and main text highlight "green dots" that don't appear in the figure. Four points are labelled with letters, but the letters aren't referred to anywhere. And it's hard to tell which point

corresponds to SMOoS.

* The authors claim that “the Mo-S1 structure always floats on the surface without sinking”. But the simulation is quite short, only 1.4 ns according to Fig. 4. The authors should perhaps claim instead that “the Mo-S1 structure remains on the surface without sinking for the duration of the 1.4 ns simulation”.

* The authors mention that “the 1T phase present in the early stages mostly disappears, indicating that MoS₂ transitions from the 1T phase to the more stable 1H phase as growth proceeds.” It’s very difficult to see this in Fig. S8. The authors should consider pointing out where the 1T phase appears in the figure, and perhaps give schematic examples of the pristine 1H and 1T phases.

* There are grammatical mistakes throughout the paper. For example, in the title, “transition metal dichalcogenides heterostructures” should be “transition metal dichalcogenide heterostructures”. In the abstract, “a highly stable intermediate easily introduces” should be “a highly stable intermediate that easily introduces”, and “Eliminating the alloying contamination... is avoiding SMMS structure” should be “Eliminating the alloying contamination involves avoiding the SMMS structure”.

(Remarks on code availability)

Version 1:

Reviewer comments:

Reviewer #1

(Remarks to the Author)

In this revision, Zhao and coworkers largely addressed inquiries and comments, as well as attempted to make appropriate changes to the manuscript. While many new details were conveyed in the revision, raised concerns about reproducibility, quality of the MLP, quality of the simulations were not satisfactorily rectified. Furthermore, with the new understanding, new questions emerged about the validity of the simulations in relation to the experimental insights (Nature 621, 499 15 (2023)), they are investigating. These concerns are paired with a remaining lack of readability/clarity of the manuscript, that despite being commented by other reviewers, has not been properly addressed. It would be advisable to put more care into the written text and scientific reporting which still appears to convey numerous mistakes. Compared to the first version, where the benefit of doubt about unknowns prevailed, their work appears now less rigor, raising doubt whether it is publishable.

Comments:

- Details on MLP training procedure which the authors added are still vague. For the final data set, the authors should at least explain the model size (e.g. number of atoms & unit cells) and the number of configurations in the training and testing set (to assess the actual number of data-points = energies & forces). The manuscript should be self-contained and it can't be expected of the reader to download the Zenodo repo to answer these questions. An exact understanding of the training set is critical, since by now it becomes clear that the training process was manually and not systematically performed.
- The various methods to create training configurations need much more detailed explanation. For example, using VASP's MLP-MD feature to get data points raises the questions, whether the data that went to training the Nequip potential were DFT calculations or VASP's MLIP prediction (which would introduce considerable noise). Also, the explanation of the Monte Carlo method implemented by the authors remains unclear and should be described in more detail. Quite confusing is the also the wording and usage of words like “epoch” that appears out-of-place and mixes with terminology of NN training (e.g. as used in Nequip). The process of drawing 100k samples from a much larger structure pool, leading to excessive redundancy of 74% also raises questions of good-practice.
- The MLP validation still falls short. From the revision it becomes apparent that no uncertainty metric was used during training (e.g. to track convergence), and instead qualitative criteria “unstable molecular dynamics” or “poor phonon spectra” called for “refining”. This should be clearly labelled as an ad-hoc approach and validation of the MLP needs to be especially meticulous. The comparison of “critical structures” is a good first step, but by far not enough to determine the MLPs accuracy over the investigated composition and structure space. A sign of significant discrepancy between MLIP and DFT is Fig. 3(j), and certainly raises doubt over the statement on page 8 “accurately reproduced DFT results”.
- The authors did not implement the previous inquiry about systematic simulations and statistics. While a more thorough data analysis is appreciated, also more test simulations (starting from different initial conditions) should be performed – which is easily achievable with the highly performant Nequip potential. But also, in the newly provided analysis, many questions arise: Why do the number of bonds start at zero in Fig. S3 and S8? What about existing bonds? Those seem to be accounted for in Fig. S12. Note also, S3 and S8 have the same caption, which is probably another mistake. Further, it might make sense to distinguish the bonds between different elements, which could help understand the growth process.
- The revised description of the simulations still requires careful refinement. It is not, yet, fully clear what is done and in the current description, terminology of ensembles is mixed up.
- The discussion of the manuscript results still requires better structure, a lot of back-and-forth is contained around Fig. 3 making it hard to follow the authors argumentation. After revision, new questions about the simulations occurred. For example, a key result is the fact, that “metal films do not float” (compared to the original experimental data in Nature 621, 499 15 (2023)), what is the significance of this finding?
- The results also contain quite a bit of inconsistent simulations: The MD simulations are conducted around deposition temperatures of 500K-1300K (?), but some analysis (e.g. mixing entropy), and other simulations (e.g. depicted in Fig. R7) are

conducted at 300K. This seems inappropriate if the synthesis mechanism is investigated. Some arguments are made about alloying where WS₂ and MoS₂ slabs appear not systematically exchanged and kinetic stabilization is argued on long MD timescales ("stability during 1.4 ns") but without a significant number of samples. These simulations thus only weakly support the authors' conclusions.

(Remarks on code availability)

Reviewer #2

(Remarks to the Author)

Report on the Revised Manuscript by Zhao et al.

1. The language and overall presentation of the manuscript have been significantly improved, making the paper much more readable and understandable. However, the abstract and the first two paragraphs of the introduction still require further polishing for clarity and conciseness.

2. The authors now clarify that they used an ideal Morse potential for the substrate simulation. Given that the study is motivated by experimental results on sapphire, it is essential that the manuscript discusses how well this idealized simulation approach approximates the actual experimental conditions. A brief justification or discussion in the main text would strengthen the manuscript.

3. The authors have added two supplementary figures to clarify the heterostructure formation. Figure S10 includes the simulation temperature, but this detail is missing in Figure S11. For clarity, the temperature should also be included in Figure S11 to clearly indicate the conditions under which the sulfurization process was studied.

Once these remaining issues are addressed, I believe the manuscript will be suitable for publication in Nature Communications.

(Remarks on code availability)

Reviewer #3

(Remarks to the Author)

(Remarks on code availability)

Reviewer #4

(Remarks to the Author)

The authors have adequately responded to my comments and I am happy to recommend the manuscript for publication.

As a small comment, I noticed that in revised Fig. S14, the figure and caption refer to a "2H" phase, but in the rest of the manuscript, this phase is referred to as "1H". I think the authors probably meant to write "1H" here.

(Remarks on code availability)

The code contained in "ocp_active.zip" is unlikely to be a usable resource for the community, as it appears to be available only in the zenodo repository as a set of python files (as opposed to a maintained repository on github), and it is unclear what files from the original Open Catalyst repository were modified or how they were modified. I don't think this should necessarily preclude publication, however, as the authors describe their methods in the manuscript and do make the code available, and more importantly they also provide their training and test structures in extxyz format, which may very well be useful for the community.

Version 2:

Reviewer comments:

Reviewer #1

(Remarks to the Author)

The authors have implemented many of the requested changes, which has significantly improved the manuscript's transparency and traceability. This removes one of the main concerns raised in the previous revision-rounds. However, I still

find that the work falls short in a few critical aspects related to reproducibility, contextualization with experimental findings (especially Ref. [23]), and the overall scientific rigor in reporting. There are also recurring issues in presentation, including inconsistent units and unclear phrasing, which are concerning at this advanced revision stage. Overall, these shortcomings present major revision points and still need careful attention.

Detailed Comments:

1) MLIP training and validation: The inclusion of the MLIP training and validation details is appreciated and now allows the reader to follow the methodology. The reported RMSEs appear acceptable considering the high-energy structures accessible at elevated temperatures. Nevertheless, to align with state-of-the-art reporting, I suggest the authors should include a measure of convergence — either RMSE versus “training round” or RMSE as a function of structure energy per atom — to assess model quality comprehensively. I also recommend revising the newly added paragraphs for clarity, as the current language is rough (especially the first one). Units for energy RMSE (meV vs. meV/atom) must be reported consistently.

2) Reproducibility: I strongly disagree with the authors’ assertion that performing multiple simulations is unnecessary. Repetition under varying initial conditions is essential, particularly for non-equilibrium simulations, to validate qualitative insights and ensure reproducibility. This is not about obtaining quantitative observables but demonstrating that observed mechanisms are robust. Performing multiple simulations could result in a brief statement such as, “All qualitative observations were confirmed in x independent runs” and would meaningfully strengthen the manuscript. Testing one’s own hypotheses in this way is fundamental to good scientific practice.

3) Temperatures in MD simulations: The selection and reporting of MD temperatures remain inconsistent. Temperatures listed in Table S1 should also appear clearly in the main text and relevant figure captions. Moreover, the rationale behind each temperature choice must be stated — currently, these choices appear arbitrary. This information is essential for readers to assess the physical relevance of the simulations.

4) Reporting of reaction energies: Certain data remain ambiguously described or mislabeled. For instance, in Figure 2, the “sinking” of Mo is attributed to DFT calculations, although the associated energy change appears to originate from MD trajectories and was only later corroborated by DFT single-point calculations according to the SI(?). The authors must clarify which quantities stem from MD and which from DFT to prevent misinterpretation, alternatively, a description of the DFT calculations is required.

5) Interpretation of results: The authors’ additional discussion connecting this work to Ref. [23] is appreciated, but some confusion persists. It would be clearer to reference this work in the new paragraph as “In the above-mentioned study [23]” rather than “A recent study reported...” to guide readers properly, after all [23] is mentioned multiple times as a motivation for this study. More importantly, the authors’ findings still seem to contradict the experimental conclusions of Ref. [23], which reported successful wafer-scale synthesis of TMDs after direct dosing of Mo on WS₂. The authors could strengthen their argument by discussing potential reconciling factors. I would think a possible interpretation are sulfur-rich conditions that might have been present during experiments in [23] and thus stabilized against the intermediate SMeMeS configuration, in accordance with the authors discussion in another context and findings around Figures S13–S16.

(Remarks on code availability)

Reviewer #3

(Remarks to the Author)

(Remarks on code availability)

Reviewer #4

(Remarks to the Author)

The authors have significantly improved the manuscript and addressed the minor concerns I noted in my previous review. I'm happy to recommend the work for publication.

(Remarks on code availability)

The code is now available on a public github repository and the changes made to the ocp package are clearer.

Version 3:

Reviewer comments:

Reviewer #1

(Remarks to the Author)

In this revision, the authors have satisfactorily addressed all remaining concerns. The validation of the MLP and the strengthened statistics of the non-equilibrium MD raise confidence in the simulations and enhance reproducibility. The contextualization with respect to reference [23] improves the overall understanding of the predictions relative to experiment. Finally, I commend the improved writing in this revision. I am pleased to recommend this work for publication now.

(Remarks on code availability)

Reviewer #3

(Remarks to the Author)

(Remarks on code availability)

Response to Reviewer 1

Comment: *In their manuscript, Zhao and coworkers conduct extensive atomistic simulations to explore the growth process of 2D transition metal dichalcogenide (TMD) van der Waals heterostructures (vdWHs). They employ a machine learning potential (MLP) based on the NequIP architecture, trained on DFT calculations, using an iterative training approach to generate the training and test sets. Non-equilibrium molecular dynamics simulations are utilized to examine the stability and growth of MoS₂/WS₂ heterostructures. The study identifies two distinct growth mechanisms: a sulfur-deficient growth via an SMeMeS (Me=W/Mo) intermediate, and a sulfur-rich growth where the MeS₂ structure remains intact while MeS_x clusters adsorb and agglomerate to form the next MeS₂ layer. The former leads to alloy formation, while the latter prevents metal mixing in the vdWHs. These findings help rationalize a recently published "two-step vapor-deposition process." Additionally, the authors propose that the SMeMeS intermediate could be used to create an atom-scale p-type Schottky contact, potentially significant for technological applications.*

The manuscript provides valuable scientific insights into the growth process and should be of significant interest to the 2D-material community. However, there are several shortcomings that require major revision. Firstly, the explanation of the training procedure lacks critical information necessary for reproducibility. Secondly, the MLP validation does not ensure the required accuracy for the results obtained. Thirdly, the non-equilibrium simulations are poorly explained and seem to lack a systematic approach, which raises questions about their robustness. Additionally, the writing is sometimes vague and difficult to follow, this applies to both the language and the organization of the results. I recommend that the authors address these issues to provide a much needed confidence in the presented results (which I believe at the moment to be insufficient) and improve readability.

Our reply: The reviewer's positive comments and valuable suggestions are highly appreciated. Regarding the main points reviewer made, we would like to respond as follows:

Comment 1:

Major comments:

1. Details of the iterative training procedure for machine learning potential are missing. I couldn't find any information on how they prepared the many different configurations in the training set – and what structures were specifically included. I believe that this also needs motivation since a repository alone leaves too many open questions. The current information on the procedures "on-the-fly MLPMD in VASP" and "Monte Carlo method" are not clear and do not offer information on structural data. Also the data-selection procedure (each iteration with 50% of new data being included) is unclear since "new" data is not defined. Note, that 50% of 100,000 configurations per iteration clearly exceeds a final data set of 26,000 entries.

Our reply: Thank you for your valuable comments. The sampling procedure of the training data and detailed explanations are shown below.

Q1: *Details of the iterative training procedure for machine learning potential are missing. I couldn't find any information on how they prepared the many different configurations in the training set?*

It starts with generating an initial dataset from quantum mechanical calculations, which is then used to train a machine learning model. Next, we validate the reliability of the model and apply it to problems to be investigated, and if it becomes problematic, e.g., unstable molecular dynamics and poor phonon spectral correspondence, fine-tune it by adding relevant DFT constitutive data. This data expansion and model retraining process are repeated iteratively until the model achieves satisfactory accuracy. The final model can then be deployed for efficient and accurate large-scale atomistic simulations.

(1) *what structures were specifically included?*

Our dataset includes:

- Single-layer and multilayer TMD structures
- MoS₂/WS₂ heterostructures
- Configurations of TMD interactions with metal clusters (Mo, W), including embedded clusters
- MoWS₂ alloy configurations
- Intermediate structures may appear during the growth process, including the SMMS type and MSMS.
- Sulfur cluster and sulfur on TMD surface

The dataset has been uploaded to the Zenodo platform (<https://doi.org/10.5281/zenodo.13777360>).

(2) *The current information on the procedures “on-the-fly MLPMD in VASP” and “Monte Carlo method” are not clear and do not offer information on structural data.*

“On-the-fly MLPMD in VASP” is an adaptive methodology developed by the VASP vendor that dynamically trains machine-learned force field during MD simulations. At each simulation step, the machine-learned force field predicts the energy and forces and determines whether the uncertainty exceeds a threshold through Bayesian error estimation. If the uncertainty exceeds a predetermined threshold, an initial DFT calculation is performed, and the resulting configuration is added to the training set to update machine-learned force field. If it is exceeded, an initial DFT calculation is performed, and the resulting configuration is added to the training set to update the machine-learned force field. This iterative process enables the machine-learned force field to learn the potential energy surface in real time, concentrating computational effort on poorly predicted regions. Consequently, this approach accelerates the exploration of the potential energy surface during the training set construction.

In our study, the “Monte Carlo (MC) method” is implemented through our self-developed procedure, where we employ a random atom-swapping strategy to rapidly perturb the system, e.g., by randomly swapping the metal atoms in the upper and lower layers of the SMMS, thus accelerating the exploration of the potential energy surface.

(3) Also the data-selection procedure (each iteration with 50% of new data being included) is unclear since “new” data is not defined. Note, that 50% of 100,000 configurations per iteration clearly exceeds a final data set of 26,000 entries.

In each iteration cycle, we employed a specialized balanced sampling strategy to construct the training dataset. Specifically:

1. **“new” data definition:** We randomly sampled 50,000 data points from newly generated configurations (some structural repetitions were inevitable during sampling), which is defined as new data.
2. Simultaneously sampled 50,000 data points from the existing training data (some structural repetitions were inevitable during sampling)
3. Combined these to form a training batch of 100,000 data points

This 1:1 sampling ratio was designed with two key objectives:

1. To ensure the model maintains accuracy on previously learned features while acquiring new ones
2. To enhance training stability by minimizing the impact of dataset size fluctuations

Since our total training set size was smaller than 100,000 configurations, some structural repetitions were inevitable during sampling. These repetitions effectively served as data weighting, aiding the model in better learning key structural features. Through this process, we ultimately obtained approximately 26,000 unique structural configurations, encompassing all critical data from the initial training to the final iteration.

During the iterative process, we employed a reduced learning rate ($1e-4$) for fine-tuning, which further ensured training stability. This combination of balanced sampling and cautious fine-tuning allowed us to effectively expand the training set's coverage while maintaining model stability and accuracy.

Accordingly, we added these discussions in the revised Supplementary Information (Page 1):

“In this study, we adopted a basic iterative training procedure that begins with generating an initial dataset from quantum mechanical calculations, which is then used to train a machine learning model. Next, we validate the reliability of the model and apply it to problems under investigation. If it becomes problematic, e.g., unstable molecular dynamics and poor phonon spectral correspondence, fine-tune it by adding relevant DFT constitutive data. This data expansion and model retraining process are repeated iteratively until the model achieves satisfactory accuracy. The final model can then be deployed for efficient and accurate large-scale atomistic simulations. Then, we employed on-the-fly molecular dynamics (MD) simulations to generate the initial training set, effectively exploring unknown configuration spaces while minimizing sampling costs. Using VASP's on-the-fly MD framework, we implemented the following parameters to optimize efficiency: a timestep of 3.5 fs ensured accurate atomic motion sampling, 20,000 maximized simulation steps, fixed unit-cell shapes, Nosé-Hoover thermostats, and a temperature range (500 K to 1300 K) covered typical growth conditions. The

ML_MB=2000 parameter constrained the number of local reference configurations in the on-the-fly MLP. Initial configurations comprised monolayer MoS₂/WS₂ systems and their combinations with homogeneous/heterogeneous metal clusters.

We expanded the training set through ten iterative epochs with distinct sampling strategies:

- **Epoch 1:** On-the-fly MD sampling of MSMS structure dynamics
- **Epoch 2:** Monte Carlo (MC) sampling of MSMS-to-SMMS transitions via random metal-sulfur exchange
- **Epoch 3:** MC sampling of alloyed SMMS intermediates through metal atom swapping
- **Epoch 4:** MC sampling of a sulfur layer on SMMS via metal-S surface atom exchange
- **Epoch 5:** Perturbed configurations near SMMS equilibrium states
- **Epoch 6:** On-the-fly MD sampling of sulfur molecular dynamics (matching initial protocol)
- **Epoch 7:** S-S/Mo-S/Mo-W dimer configurations at varied distances
- **Epochs 8–17:** MLPMD simulations of SMMS growth and Mo deposition (key structures extracted every 10,000 steps)

Iterative model refinement incorporated both historical and new data. We randomly sampled 50,000 data points from newly generated configurations (with some structural repetitions being inevitable during sampling), which we defined as the new data. During each epoch, we constructed a training set of 100,000 configurations through 1:1 random sampling of historical data (cumulative from prior epochs) and newly generated data. The MLP fine-tuning employed a learning rate of 1×10^{-4} to balance convergence speed and accuracy. The final dataset contained approximately 26,000 configurations from initial training and all epochs, ensuring structural diversity and representativeness critical for robust MLP development. The dataset has been uploaded to the Zenodo repository (<https://doi.org/10.5281/zenodo.13777360>).”

Comment 2: *The MLP validation consists of fairly meaningless RMSE's and some MD simulations validating(?) the MLP and AIMD validating the stability predictions. These calculations have limited certainty of the accuracy. What I believe is much more important is, e.g. comparing the predicted relative stability of optimized and non-optimized structures (e.g. SMeMeS vs. MeS₂) via the MLP vs. DFT.*

Our reply: Many thanks for the reviewer's thorough review and insightful comments. In response to the comprehensiveness of MLP validation, we added the comparative analysis of the relative stabilities of critical configurations. By systematically comparing the energy predictions of optimized and non-optimized structures obtained from MLP and DFT (see the details in Table S1 shown below), it can be seen that a high consistency between MLP and DFT in the relative stabilities of key configurations, such as SMMS and MS₂. Specifically, for the SMMS-like configuration generated by direct molecular dynamics deposition and the optimized SMMS structure, the energy differences are 0.080 eV and 0.068 eV, obtained from DFT calculations and MLP predictions, respectively. Similarly, for the SMoWS system, the corresponding energy

differences are 0.090 eV from DFT simulation and 0.077 eV from MLP prediction. These results demonstrate that our MLP can accurately compute the energy differences between different structures and reliably describe the relative stabilities of materials.

To increase our manuscript's readability, we added these discussions in the revision (page 5) and a new table (S1) in the revised Supplementary Information (page 4):

“By systematically comparing the energy predictions of key structures obtained from MLP and DFT shown in Table S1, it can be seen that a high consistency between MLP and DFT in the relative stabilities of key configurations such as SMMS and MS₂. Specifically, for the SMMS-like configuration generated by direct MD deposition and the constructed and optimized SMMS structure, the energy difference is 0.080 eV and 0.068 eV, obtained from DFT calculations and MLP predictions, respectively. Similarly, for the SMoWS system, the corresponding energy difference is 0.090 eV from DFT simulation and 0.077 eV from MLP prediction. These results demonstrate that our MLP can accurately compute the energy differences between different structures and reliably describe the relative stabilities of materials.”

“Table S1. Comparison of energies (eV/atom) for critical structures calculated by DFT and MLP.

Structure	DFT (eV/atom)	MLP (eV/atom)	Δ E (eV/atom)
S solid	-4.323	-4.311	-0.013
1H-MoS ₂	-7.582	-7.583	0.001
1T-MoS ₂	-7.298	-7.300	0.002
1T'-MoS ₂	-7.390	-7.380	-0.010
1H-WS ₂	-8.214	-8.144	-0.070
1T-WS ₂	-7.893	-7.899	0.005
1T'-WS ₂	-8.026	-8.017	-0.009
MoSWS	-8.375	-8.416	0.041
MoSMoS	-7.978	-8.007	0.029
SMoMoS	-8.409	-8.382	-0.027
SMoWS	-8.895	-8.892	-0.002
SMMS	-8.897	-8.895	-0.002
SMoMoS (from MD)	-8.329	-8.314	-0.015
SMoWS (from MD)	-8.804	-8.815	0.011

”

Comment 3: *The non-equilibrium MD simulations are also unclear. How exactly are new species introduced during the MD simulations? Why are they introduced with low velocities. Furthermore, why didn't the authors collect statistics and clearly quantify e.g. mean times of incorporation. The efficient MLP would certainly allow for this. Currently, all results are qualitative observations without any quantitative indication.*

Our reply: Many thanks for the reviewer's thorough review and insightful comments.

(1) *How exactly are new species introduced during the MD simulations?*

The atomic deposition was achieved using the fix deposit command in LAMMPS, which introduces a Mo atom with a downward velocity of 5 Å/ps at random positions within the top region of the simulation box every 6000 steps. The initial kinetic energy of deposited atoms was set to 0.12 eV. Regarding the implementation details of non-equilibrium MD simulations, we have provided an expanded description in the Methods section (page 18):

“The atomic deposition was achieved through the fix deposit command in LAMMPS, which introduces a Mo atom with downward velocity of 5 Å/ps at random positions within the top region of the simulation box every 6000 steps. The initial kinetic energy of deposited atoms was set to 0.12 eV.”

(2) *Why are they introduced with low velocities?*

In non-equilibrium MD simulations, a lower kinetic energy was chosen to show the intrinsic nature of the embedding behavior, as a better representation of the embedding behavior does not necessitate a high kinetic energy. As shown in Fig. R1, we simulated cases with 1 eV and 5 eV of incident kinetic energy, exhibiting similar embedding behavior.

Fig. R1: (a) Simulated 1 eV kinetic energy deposition of a single Mo layer at 900 K on MoS₂. (b) Simulated 5 eV kinetic energy deposition of a single Mo layer at 900 K on MoS₂.

Relevant discussions and a new figure (Fig. S4) are added in the revised main text (page 7) and Supplementary Information:

“As shown in Fig. S4, when Mo atoms were deposited at higher kinetic energies, such as 1 eV and 5 eV, they remain embedded in MoS₂. This indicates that the embedding behavior of Mo atoms is not sensitive to kinetic energy.”

(1) *why didn't the authors collect statistics and clearly quantify e.g. mean times of incorporation?*

We thank the reviewer for the valuable comments. Quantitative data are essential for understanding the MD process. Therefore, in the revised manuscript, we systematically quantified the evolution of metal-metal and metal-sulfur bond counts [Fig. R2, Fig. R3 and Fig. R4]. These quantitative data reveal the atomic incorporation process and sulfurization reaction kinetics.

Specifically, the progressive increase in metal-metal bonds indicates the continuous embedding of deposited atoms. The sudden surge of the metal-sulfur bonds during sulfurization corresponds to a critical stage of coordination reconstruction. These quantitative analyses provide crucial evidence for understanding the formation mechanism of intermediate structures.

Fig. R2: Number of Mo/W-Mo/W bonds (including Mo-Mo, Mo-W, and W-W bonds) during Mo deposition on a WS₂ substrate. A bond cutoff distance of 2.85 Å was applied for bond identification in both cases.

Fig. R3: Number of Mo/W-Mo/W bonds (including Mo-Mo, Mo-W, and W-W bonds) during Mo deposition on a WS₂ substrate. A bond cutoff distance of 2.85 Å was applied for bond identification in both cases.

Fig. R4: (a) Evolution of the number of Mo-S bonds during sulfur deposition on the SMOsMoS intermediate structure. Excess sulfur atoms extract Mo atoms from the SMOsMoS structure, leading to the formation of a bilayer MoS₂ heterostructure. (c) Evolution of the number of Mo/W-S bonds during sulfur deposition on the alloyed S(Mo_{0.5}W_{0.5})(W_{0.5}Mo_{0.5})S structure. Sulfur atoms facilitate simultaneous extraction of both Mo and W atoms, ultimately forming an alloyed bilayer Mo_{0.5}W_{0.5}S₂ configuration. A cutoff distance of 2.6 Å was applied for both Mo-S and W-S bond identifications.

Three new figures (Fig. S3, Fig. S8 and Fig. S12) were added in the Supplementary Information. Relevant discussions were added in the revised main text:

On Page 7,

“Fig. S3 depicts the evolution of Mo-Mo bonds during continuous deposition, indirectly reflecting the embedding process. We observed that the number of Mo-Mo bonds gradually increased with continued deposition and reached saturation once deposition ceased.”

On page 9,

“Fig. S8 shows the evolution of Mo/W–Mo/W bonds during continuous Mo deposition, indirectly reflecting the degree of Mo atom insertion. The number of these bonds gradually increases with deposition and saturates once deposition ceases.”

On page 12,

“As shown in Fig. S12(a), there are very few initial Mo-S bonds because the S atoms deposited on the SMMS surface cannot bond with Mo. As the MD simulation progresses, some Mo atoms are pulled to the surface to bond with the deposited S atoms, leading to an increase in the number of Mo-S bonds. Between 700 and 1000 ps, a large number of Mo atoms are pulled to the surface, resulting in a substantial increase in Mo-S bonds. The number of Mo-S bonds tends to be saturated after about 1.2 ns, when a bilayer MoS₂ is formed.”

On page 13,

“Therefore, the resulting structure was not a distinct MoS₂/WS₂ vdWHs but rather an alloyed Mo_xW_{1-x}S₂/Mo_{1-x}W_xS₂ vdWHs. As shown in Fig. S12(b), the evolution of Mo/W-S bonds clearly illustrates this process.”

Comment 4: *The discussion on of the growth mechanism should be more organized. Each paragraph appears as a loose description of what simulation was performed and what was seen lacking motivation that would help the reader gaining the very important scientific insight.*

Our reply: We thank the reviewer for the thorough review and insightful comments. In the revised manuscript, we have systematically restructured the mechanistic analysis framework to enhance logical flow and scientific clarity. The discussion now progresses through a well-defined hierarchy of fundamental processes: First, we employ molecular dynamics simulations to reveal the intrinsic tendency of metal atoms to spontaneously embed beneath the sulfur layer during initial deposition, forming subsurface metal-bonded configurations. This is followed by comprehensive thermodynamic calculations and phonon dispersion analysis that establish the SMMS intermediate as a structurally stable transition state. Next, simulations of hetero-deposition processes elucidate the thermodynamic preference and stabilization mechanisms for alloyed SMMS formation. Non-equilibrium MD simulations then map the dynamic evolution pathways of these intermediates during sulfurization, while comparative deposition studies demonstrate how co-deposition of Mo-S clusters effectively suppresses SMMS formation. Finally, electronic structure calculations unveil the unique advantages of these intermediates for device applications. To improve navigability, we have introduced descriptive subheadings to guide readers through each key mechanistic stage.

Comment 5: *In illustrate the latter point, data from various simulations appears sometimes erratically combined. An example is the discussion of the stability of the bilayer system. If I understood correctly, the structure for MoSMoS, SMOMoS, SMO₃S, SMO₄S (Fig2. (f-i)) are obtained by geometry optimization, not by MD simulation. Did the authors observe the SMOMoS (Fig.2 (g)) structure during the MD simulation starting from the MoSMoS structure (Fig.2 (f))? If not, the statement of “all deposited Mo atoms spontaneously sank into the MoS₂ layer, tending to form the intriguing SMOMoS structure, as shown in Fig. 2(g).” and “In addition to the SMOMoS structure formed by depositing 1 ML of Mo atoms” might be misleading, because as seen in Fig. S5 (b) it is less likely that the formation of SMOMoS structure from MoSMoS structure happens.*

Also, in this discussion, the authors showed a lot of data:

- *energy landscape of MoSMoS, SMOMoS, SMO₃S, SMO₄S structures (Fig2. (f-i))*
- *phonon band structures (Fig2. (l))*
- *tendency that Mo atoms sink beneath the S layer during the MLP-MD simulation (Fig2. (e))*
- *tendency that Mo atoms sink beneath the S layer during the AIMD simulation (Fig S5(b))*
- *stability of SMOMoS during the MLP-MD simulation (Fig S3 (a)(d))*

○ *stability of SMoMoS during the AIMD simulation (Fig S5 (c))*

As such, the data from geometry optimization and MD simulations using DFT and MLP are given, and it is hard to understand how they complement each other in the current version of the manuscript. This should be more clarified for better discussions.

Our reply: We sincerely appreciate the reviewer's thorough critique. The original description regarding the formation mechanism of the SMoMoS structure indeed requires clarification. The four configurations shown in Fig. 2(f-i) were obtained through DFT geometry optimizations, rather than directly from MD simulation trajectories. The key evidence for the SMoMoS formation pathway originates from MLPMD simulations, which reveal that Mo atoms spontaneously migrate into the MoS₂ layer when directly deposited on its surface. MLPMD simulations of sequential Mo deposition suggest that SMoMoS serves as a naturally formed SMMS (S-M-M-S) intermediate during metal atom deposition. Crucially, the MoSMoS configuration was never observed as a transition state in simulations. Formation energy calculations further indicate its high-energy state, confirming negligible likelihood of existing as a metastable phase. We acknowledge that the statement of “all deposited Mo atoms spontaneously sank into the MoS₂ layer, tending to form the intriguing SMoMoS structure, as shown in Fig. 2(g).” and “In addition to the SMoMoS structure formed by depositing 1 ML of Mo atoms” might be misleading. We have revised the relevant discussion in the manuscript. Throughout the MLPMD simulation, no Mo atoms formed the MoSMoS structure [Fig. R5(f)]. Instead, all deposited Mo atoms spontaneously sank into the MoS₂ layer. These findings suggest that under practical deposition conditions, Mo atoms preferentially form an SMoMoS structure [Fig. R5(g)]. This SMoMoS structure was then constructed and optimized by DFT. The results show this SMoMoS structure exhibits an energy 0.080 eV/atom lower than that of the directly formed embedded structure from MD simulations [Fig. R5(e)].

To reveal the dynamic behavior during the actual growth process, the MLP-MD simulations (Fig. R5(e)) demonstrate the deposition dynamics of metal atoms and the pathways of structural evolution on a nanosecond timescale. Motivated by these simulation insights, we systematically constructed and optimized the structures (Fig. R5(f-i)). Fig. R5(j) shows the relative thermodynamic stability of these structures. Additionally, the phonon spectrum analysis (Fig. R5(l)) further validates the dynamical stability of SMoMoS, ruling out the possibility of imaginary frequencies. Long-term MLPMD simulations (5.5 ns) further validate the structural robustness of SMoMoS under thermal fluctuations. Furthermore, a shorter period of AIMD simulation was conducted to verify the reliability of the embedding behavior and the stability of the SMMS structure, serving as a validation and complement to the MLP results.

Fig. R5: (a) The structure of Mo atoms deposited and (b) embedded into the MoS₂ layer, (c) along with the corresponding energy plots (blue line represents raw energy, while red line shows the low-pass-filtered result). (d-e) Snapshots of 0.25 monolayer (ML) (d) and 1.0 ML (e) Mo atom deposition simulated at 900 K on the existing MoS₂ layer. (f-i) Four possible structures: (f) MoSMoS, (g) SMoMoS, (h) SMo₃S, and (i) SMo₄S with more Mo atoms embedded. (j) Formation energy convex hull of the considered MoS structures, with green dots corresponding to energies of structures found in the 2D Materials Database (MatHub-2d). (k) The unit cell structure of SMoMoS and (l) its phonon dispersion relation.

We have modified these discussions in the revised main text (page 7):

“

Throughout the MLPMD simulation, no Mo atoms formed the MoSMoS structure [Fig. 2(f)]. Instead, all deposited Mo atoms spontaneously sank into the MoS₂ layer. These findings suggest that under practical deposition conditions, Mo atoms preferentially form an SMoMoS structure [Fig. 2(g)]. This SMoMoS structure was then constructed and optimized using DFT, revealing that it exhibits an energy 0.080 eV/atom lower than that of the directly formed embedded structure from MD simulations [Fig. 2(e)].

We conducted a detailed comparison of these two structures in terms of energy and dynamic stability. The energy significantly decreased by 1.73 eV per Mo atom when transitioning from MoSMoS to SMoMoS, indicating a strong driving force towards the formation of the SMoMoS structure. In addition to the SMoMoS structure formed by depositing 1 ML of Mo atoms, we constructed and optimized structures such as SMo₃S [Fig. 2(h)] and SMo₄S [Fig. 2(i)]., which may form with an increased number of Mo atoms. The convex hull of MoS compounds was plotted by varying the Mo ratio, referencing the bulk phases of elements Mo and S, as shown in Fig. 2(j). Notably, MoS₂ has the lowest energy, while SMoMoS is situated 196 meV/atom above

the convex hull. We also found structures that match this elemental ratio from other 2D materials databases (MatHub-2d)³⁷[Fig. S5], with their formation energies indicated by green dots in the figure, showing that they are 253 meV/atom higher than the S₂MoMoS structure. S₂Mo₃S and S₂Mo₄S are 221 meV/atom and 218 meV/atom above the convex hull, respectively, indicating that when approximately 1 ML of Mo atoms is deposited on the MoS₂ surface, S₂MoMoS is the most likely intermediate. The phonon dispersion of the S₂MoMoS structure showed no imaginary frequencies [Fig. 2(k) and Fig. 2(l)], further confirming that MoS₂MoS would spontaneously transform into S₂MoMoS. It is worth noting that the phonon dispersion simulated by the MLP accurately reproduced the DFT results, validating the accuracy of our MLP in describing structural stability. Long-term MLPMD simulations further validated the stability of S₂MoMoS, as shown in Fig. S6.

Furthermore, we used DFT to validate the key processes described above. Fig. S7(a) illustrates the process of a single Mo atom into the MoS₂ layer. Despite the relatively low initial kinetic energy, the Mo atom quickly embedded into the MoS₂ layer within approximately 700 fs, as indicated by the sudden drop in energy. Notably, compared to pre-adsorbed Mo atoms, vacuum-deposited Mo atoms significantly shorten the embedding time to about 700 fs — much shorter than the approximately 50 ps required for surface-adsorbed atoms in previous simulations. Fig. S7(b) expands our investigation of the behavior of the Mo atomic layer on MoS₂. As the Mo atoms gradually embed into the MoS₂ structure, they descend accordingly. This energy decrease is significantly lower than that of the initial configuration, further confirming the thermodynamic favorability of Mo atom embedding. Fig. S7(c) focuses on the stability of the formed S₂MoMoS structure over approximately 10 ps.

”

Comment 6:

Minor comments:

● *Please check the numbering for the figures. For example, Fig. S4 comes earlier than Fig. S3, which is not kind to readers.*

Our reply: Thank you for your careful review of our manuscript and the valuable comments. We have repositioned all supplementary figures to ensure their numbering is consistent with the references in the main text.

Comment 7: ● *Small comments on the figures:*

○ *Fig.2 (j) it is stated that “with green dots” but I couldn't find the green point*

○ *Fig.2 (c) what is the red line and blue line?*

○ *For the caption for Fig.2, the labels (a) to (e) come after the caption but (f) to (l) come before the caption, which is, I think, very confusing.*

○ *Fig.5 (b) SMOWS should be S₂MoWS*

Our reply: Thank you for your detailed and constructive feedback, which has significantly enhanced the clarity and rigor of our manuscript. We have implemented systematic revisions to address the issues related to the figures and ensure the accuracy of data presentation. We have redrawn the convex hull plot to better highlight the green dots [Fig. R8]. For Fig. 2(c), we added clarifications in the caption: the blue line represents the original energy data, while the red line depicts the trajectory after low-pass filtering. This filtering process effectively suppresses high-frequency thermal noise, providing a more intuitive representation of the energy transitions during the atomic deposition process.

Fig. R6: Revised Fig. 2(j).

Comment 8: • For the phonon band structures, the data from DFT and MLP are plotted together on the same figure, but I suggest to show only DFT results. The authors stated in page 3 “It is worth noting that the phonon dispersion simulated by the MLP accurately reproduced the DFT results, validating the accuracy of our MLP in terms of atomic forces.”, but some degeneracy at the gamma point are not well described by MLP, and it is hard to tell the accuracy of MLP from this figure. For Fig.3 (j-l), the authors don’t mention the differences between DFT and MLP, so I suggest removing the phonon band structures by MLP for better clarity.

Our reply: We sincerely thank the reviewer for the valuable suggestion. The MLP developed in this study is primarily aimed at dynamic simulations of growth processes. Although there are slight differences in the description of phonon degeneracy near the Γ point by the MLP, this does not affect its reliability in simulating non-equilibrium processes such as atomic deposition and structural reorganization. Furthermore, retaining the MLP results can demonstrate the reliability of the MLP in describing structural stability. Therefore, we believe that the MLP portion should be retained.

Comment 9: • Equations in Page 4

- Please define N and T .
- E_{SMoWS} is energy, then why is E formation energy?
- x should be “one Mo atom” not “one metal atom”?

Our reply: We sincerely thank the reviewer for the valuable suggestion. In this equation, N represents the total number of atoms in the system, T is the thermodynamic temperature, E_{SMoWS} denotes the energy of the unalloyed SMMS structure, E represents the energy of the alloyed configuration, and x is the fraction of Mo atoms in the upper metal layer.

We have added these descriptions in the revised main text (page 10):

“To determine the thermodynamic driving force for alloying, we calculated the free energy change ΔF for varying distributions of metal atoms between the upper and lower layers. Here, ΔF is defined as

$$\Delta F = \frac{1}{N} [E - E_{SMoWS} - T\Delta S],$$

where E_{SMoWS} denotes the energy of the unalloyed SMMS structure [Fig. 3(h)], E is the energy of the alloyed configuration, T is the temperature (300 K), and ΔS represents the mixing entropy for Mo/W alloying. The mixing entropy S is given by

$$S = (N_{Mo} + N_W)k_b[-x\ln(x) - (1-x)\ln(1-x)],$$

where k_b is the Boltzmann constant, x is the fraction of Mo atoms in the upper metal layer, and N_{Mo} and N_W are the numbers of Mo and W atoms, respectively.

”

Comment 10: • Page 5: “Throughout the calculations, general precision was used”, it would be better to specify the precision. “general precision” is vague.

Our reply: We appreciate the reviewer's valuable feedback. The phrase "general precision" in the original text was indeed misleading and has now been removed. The details of the precision can be found in the Methods part.

Comment 11: • For the simulation given in video S4, the S layer looks floating above the SMoMoS layer. Is it constrained?

Our reply: In the referenced simulation (Video S4), no positional constraints were applied to any atomic layers, including the sulfur layer. The molecular dynamics simulations were conducted under fully unconstrained NVT ensemble conditions, allowing all atoms to move freely according to the interatomic forces described by MLP.

Comment 12: • Page 5, what do you mean by “a specific rate of 516 ps per layer”?

Our reply: We thank the reviewer for their valuable comment. The phrase has been revised to clarify that the deposition of the second layer requires a duration of 516 ps.

We have this description in the revised main text (page 14):

“The Mo and S atoms required for the growth of one layer of MoS₂ were deposited onto the surface within 516 ps, after which the deposition was stopped.”

Comment 13: • *For most of the systems, the system size is not clearly stated but it should be given.*

Our reply: We sincerely thank the reviewer for the constructive suggestion. In response, we have added detailed system size parameters in the "MD" section of the Methods. Specifically, we have included the dimensions of the $4\sqrt{3} \times 7$ supercell used for Mo atom deposition simulations, the system sizes for SMOoS and SMMS sulfurized process simulations, and the diffusion behavior of Mo-S clusters on the MoS₂ surface. The $6\sqrt{3} \times 8$ supercell employed for studying the growth of the second MoS₂ layer.

We have this description in the revised main text (page 18):

“For Mo atom deposition, we used a $4\sqrt{3} \times 7$ MoS₂ or WS₂ supercell at 900 K. The SMOoS and SMMS sulfurized process simulations were performed using a $4\sqrt{3} \times 7$ supercell at 1100 K. The SMOoS and SMMS sulfurized process simulations were performed using a $4\sqrt{3} \times 7$ supercell at 1100 K. The diffusion behavior of Mo-S clusters on the MoS₂ surface was also investigated by $4\sqrt{3} \times 7$ MoS₂ supercell at 1100 K. For the simultaneous Mo and S deposition to grow the second MoS₂ layer, we employed a larger $6\sqrt{3} \times 8$ MoS₂ or WS₂ supercell at 1100 K.”

Comment 14: • *Fig. S5(c) shows the AIMD simulation of SMOoS structure until 10 ps, but what happens after 10 ps? Did the system retain the SMOoS structure?*

Our reply: We thank the reviewer for raising this important question. Yes, the system retains the SMOoS structure after 10 ps. Due to computational resource limitations, our AIMD simulation was indeed restricted to a short duration of 10 ps. However, during this simulation, no significant structural degradation or instability was observed in the SMOoS configuration. To comprehensively assess its thermodynamic stability, we conducted an extended simulation using MLPMD for a duration of 5.5 ns. The results demonstrate that the intermediate structure maintains its intact lattice configuration throughout the simulation, further confirming its thermodynamic stability as a growth intermediate.

Comment 15: • *Again, are the structures of Fig.3 (g-i) obtained by geometry optimization and not appeared during the MD simulations?*

Our reply: We appreciate the reviewer's careful observation. The structures shown in Fig. 3(g-i) were indeed constructed through geometry optimization as theoretical models and did not appear during the MD simulations. To eliminate any confusion between these theoretical models and the dynamic evolution results observed in the MD simulations, we have added clarifying remarks in the main text.

We added this description in the revised main text (page 10):

“Due to the similar interaction between Mo atoms and WS₂, as previously observed with MoS₂, the constructed SMOoS intermediate structure shown in Fig. 3(h) emerges as another crucial

configuration to consider. Its energy is 0.090 eV/atom lower than that of the directly formed structure from MD simulations [Fig. 3(c)].”

Response to Reviewer 2

Comment: *The manuscript by Zhao et al. presents a machine learning study investigating the atomistic details of van der Waals heterostructure (vdWH) formation during chemical synthesis. While the topic is inherently interesting and relevant, the paper does not meet the necessary criteria for publication due to several significant shortcomings.*

Our reply: Thank you for your thoughtful review and detailed feedback on our manuscript. We appreciate your interest in our study on the atomistic details of van der Waals heterostructure formation during chemical synthesis. We seriously considered each point raised and addressed them comprehensively in our revised manuscript. We are committed to improving the manuscript and believe that the revisions will significantly enhance its quality and clarity. We welcome any additional comments or specific suggestions that could guide us further in addressing these concerns. Below are our point-by-point responses:

Comment 1: *First, the language throughout the manuscript, particularly in the first two pages, is poorly constructed, making it difficult—at times even impossible—to discern the authors' intended meaning. This lack of clarity severely impacts the readability and comprehension of the work.*

Our reply: Thank you for the reviewer's suggestion. To address this comment, we meticulously revised the language throughout the manuscript, with a particular focus on the first two pages, to ensure that our intended meaning is clear and easily discernible. To improve the overall quality of the writing, we utilized a professional editing service to assist in refining the language and structure.

Comment 2: *Second, the overall structure and logic of the manuscript are unclear. The progression of ideas is disorganized, making it challenging to follow the content from beginning to end. Additionally, several critical details are omitted, which undermines the rigor of the study. For instance:*

- (i) The temperatures at which the calculations were conducted are not clearly specified.*
- (ii) The substrates used for investigating the formation of vdWHs are not identified.*

Our reply: Thank you for the reviewer's thorough review and insightful comments.

In the revised manuscript, we have systematically restructured the mechanistic analysis framework to enhance logical flow and scientific clarity. The discussion now progresses through a well-defined hierarchy of fundamental processes: First, we employ molecular dynamics simulations to reveal the intrinsic tendency of metal atoms to spontaneously embed beneath the sulfur layer during initial deposition, forming subsurface metal-bonded configurations. This is followed by comprehensive thermodynamic calculations and phonon dispersion analysis that establish the SMMS intermediate as a structurally stable transition state. Next, simulations of hetero-deposition processes elucidate the thermodynamic preference and stabilization mechanisms

for alloyed SMMS formation. Non-equilibrium MD simulations then map the dynamic evolution pathways of these intermediates during sulfurization, while comparative deposition studies demonstrate how co-deposition of Mo-S clusters effectively suppresses SMMS formation. Finally, electronic structure calculations unveil the unique advantages of these intermediates for device applications. To improve navigability, we have introduced descriptive subheadings to guide readers through each key mechanistic stage.

For the omitted critical details:

(i) The temperature settings for all MD simulations in this study were as follows: the Mo atomic deposition process was carried out at 900 K, while the simulation of the SMoMoS and SMMS sulfurized processes, the diffusion behavior of Mo-S clusters on the MoS₂ surface, and the simulation of the co-deposition of Mo and S for the growth of a second MoS₂ layer were performed at 1100 K.

(ii) The simulations employed an ideal Morse substrate ($V(r) = D_0(1 - e^{-\alpha(r-r_0)})$) with parameters $D_0 = 0.2$ eV, $\alpha = 1.5$, and $r_0 = 3.5$ Å.

We have below description in the revised main text (page 18):

“The simulations employed an ideal Morse substrate ($V(r) = D_0(1 - e^{-\alpha(r-r_0)})$) with parameters $D_0 = 0.2$ eV, $\alpha = 1.5$, and $r_0 = 3.5$ Å. For Mo atom deposition, we used a $4\sqrt{3} \times 7$ MoS₂ or WS₂ supercell at 900 K. The SMoMoS and SMMS sulfurized process simulations were performed using a $4\sqrt{3} \times 7$ supercell at 1100 K. The diffusion behavior of Mo-S clusters on the MoS₂ surface was also investigated by $4\sqrt{3} \times 7$ MoS₂ supercell at 1100 K. For the simultaneous Mo and S deposition to grow the second MoS₂ layer, we employed a larger $6\sqrt{3} \times 8$ MoS₂ or WS₂ supercell at 1100 K.”

Comment 3: *The authors motivate their theoretical study with experimental results published in Nature (621, 499, 2023). However, they fail to provide sufficient a meaningful comparison with their theoretical results with the experiment. Such a comparison is essential for evaluating the reliability of the study and its implications for the synthesis of pure vdWHs (without alloying).*

In its current state, the manuscript lacks the clarity, structure, and critical details necessary for a proper review. Given these substantial deficiencies, I regret to conclude that the manuscript cannot be considered for publication and must be rejected.

Our reply: We sincerely thank the reviewer for their thorough review and insightful comments. We acknowledge the importance of validating theoretical results against experimental observations. The primary objective of our study is to elucidate the intermediate structures during the two-step growth process. We have identified the SMMS structure as a crucial intermediate state that is essential for understanding the two-step growth mechanism. However, directly observing these intermediate states experimentally is highly challenging. While the thermodynamically stable SMMS structure can lead to alloyed TMD vdWHs, which contradicts the non-alloyed vdWHs obtained in experiments. It is important to recognize that experimental synthesis is a complex kinetic process, and the outcomes are not always the thermodynamic ground state. Therefore, we have conducted further investigations to address this discrepancy.

Fig. R7: (a-e) Snapshots of the growth structure of MoS₂/WS₂ vdWHs during the two-step vapor-deposition process (Mo atoms are deposited on WS₂) in 300K.

As shown in Fig. R9, we observed that Mo atoms continued to embed during the deposition process simulated at 300 K. No Mo-W swap phenomenon was observed after the embedding occurred. This suggests that at lower temperatures, the intermediate state is likely formed to be SMoWS, rather than the thermodynamically stable alloyed SMMS.

Fig. R8: (a-e) MLPMD simulation with a large number of S atoms placed on top of the alloyed SMoWS structure. The S atoms pull out Mo atoms, forming a MoS₂/WS₂ vdWHs structure.

As shown in Fig. R10, we subsequently simulated the sulfurization process of the SMoWS structure. The simulation results demonstrate that during sulfurization, the Mo atoms in the upper layer detach from the SMoWS structure, while S atoms rapidly occupy these positions, thereby preventing the extraction of W atoms from the lower layer. After sulfurization, the SMoWS structure forms a non-alloyed MoS₂ layer in the upper layer, which is consistent with experimental observations. This mechanism provides an atomic-level explanation for the non-alloyed growth observed in experiments.

Relevant new figures (Fig. S10 and Fig. S11(f-j)) have been added to the Supplementary Information. We have added these discussions in the revised main text,

On page 11,

“

Although the alloyed structure is the thermodynamic ground state, the actual growth process may be influenced by kinetic factors. We further investigated the deposition behavior of Mo atoms on the WS₂ surface at 300 K. As shown in Fig. S10, during the deposition process, Mo atoms continued to embed into WS₂ to form the SMoWS structure, but no Mo-W atomic swap was observed. In the subsequent 4.5 ns simulation, no alloying occurred. This indicates that the formation of the alloyed SMMS intermediate state is constrained by kinetic barriers at lower temperatures. Therefore, under lower temperature deposition conditions, the SMoWS structure or a low-alloyed SMMS intermediate state is predominantly formed.

”

On page 13,

“

According to the experimental results of Zhou et al.²³, the successful synthesis of non-alloyed MoS₂/WS₂ heterostructures suggests that the formation of the SMMS intermediate state may have been avoided through process control during the experiment. In the scenario where the intermediate state is the SMoWS structure, we further conducted sulfurization MLPMD simulations. As shown in Fig. S11(f-j), during the simulation, the upper Mo atoms detached from the SMoWS structure. S atoms quickly occupied the original positions of the Mo atoms, effectively preventing the lower W atoms from being extracted. Following the sulfurization of SMoWS, a non-alloyed MoS₂ layer formed on the upper layer. This mechanism provides an atomic-level explanation for the experimentally observed non-alloyed growth.

”

Response to Reviewer 3

Comment: *I co-reviewed this manuscript with one of the reviewers who provided the listed reports. This is part of the Nature Communications initiative to facilitate training in peer review and to provide appropriate recognition for Early Career Researchers who co-review manuscripts.*

Our reply: We sincerely appreciate your notification regarding the co-reviewing arrangement. We fully understand and strongly support Nature Communications' initiative in fostering peer review training and providing appropriate recognition for Early Career Researchers. We have carefully considered all comments from the reviewers and have integrated these valuable suggestions into our comprehensive revision. Each point raised has been thoroughly addressed in our detailed response, and corresponding modifications have been made throughout the manuscript to improve its quality and clarity. We believe this collaborative review process has significantly enhanced our work, and we are grateful for the constructive feedback from all reviewers.

Response to Reviewer 4

Comment: *The authors have written an interesting paper on the formation of a stable intermediate state which they call SMMS that forms when depositing bare Mo atoms on MoS₂ or WS₂. While there isn't much methodological innovation in the paper, at least as currently presented, it is a nice and fairly complex application of ML potentials to an important class of materials that sheds some light on the formation of MoS₂/WS₂ bilayers, namely that Mo must be deposited with sulfur to ensure that Mo doesn't sink below the surface.*

I do have some concerns that I think should be addressed if the paper is to be published in Nature Communications.

Our reply: Thank you for the reviewer's detailed review of our paper and the reviewer's valuable suggestions. We greatly appreciate the comments raised by the reviewer, which will help us further improve the quality and clarity of our work.

Comment 1:

Major concerns

** The iterative learning approach presented in the paper is unclear. The results section states that the initial training set was generated with on-the-fly learning and then "continuously enriched with new structures throughout the iterations". The methods and SI do not provide much additional information, beyond the number of iterations (17) and the number of "data points" per iteration (100,000). Are these data points entire structures or individual forces and energies? How exactly were these new structures generated? The SI alludes to a Monte Carlo method, but this is vague. In my opinion, the generation of the training set is central to the paper and will be of interest to others in the field. I suggest the authors describe their method in much more detail. As presented currently, it is not possible to reproduce it.*

Our reply: Thank you for the reviewer's thorough review and insightful comments. We agree that providing more detailed information about our methods is essential for ensuring the reproducibility of our results. Following the reviewer's suggestions, we have revised our manuscript accordingly.

(1) The iterative learning approach presented in the paper is unclear. The results section states that the initial training set was generated with on-the-fly learning and then "continuously enriched with new structures throughout the iterations". The methods and SI do not provide much additional information, beyond the number of iterations (17) and the number of "data points" per iteration (100,000).

Here, we provide a detailed explanation of our approach:

The process begins with generating an initial dataset from quantum mechanical calculations, which is then used to train a machine learning model. Next, we validate the reliability of the model, apply it to problems to be investigated, and if it becomes problematic, e.g., unstable molecular dynamics and poor phonon spectral correspondence, fine-tune it by adding relevant DFT constitutive data. This data expansion and model retraining process are repeated iteratively until the model achieves satisfactory accuracy. The final model can then be deployed for efficient and accurate large-scale atomistic simulations.

In each iteration cycle, we employed a specialized balanced sampling strategy to construct the training dataset. Specifically:

1. We randomly sampled 50,000 data points from newly generated configurations (with some structural repetitions being inevitable during sampling)
2. Simultaneously, we sampled another 50,000 data points from the existing training data (again, some structural repetitions were inevitable during sampling)
3. Combined these samples to form a training batch consisting of 100,000 data points

This 1:1 sampling ratio was designed with two key objectives:

1. To ensure the model maintains accuracy on previously learned features while acquiring new ones
2. To enhance training stability by minimizing the impact of dataset size fluctuations

Given that our total training set size was smaller than 100,000 configurations, some structural repetitions were inevitable during sampling. These repetitions effectively served as data weighting, helping the model better learn key structural features. Through this process, we ultimately obtained 26,000 unique structural configurations that encompass all critical data from initial training to final iteration.

(2) Are these data points entire structures or individual forces and energies?

The “data points” refer to an atomic structure and its corresponding total energy and force per atom.

(3) How exactly were these new structures generated? The SI alludes to a Monte Carlo method, but this is vague. In my opinion, the generation of the training set is central to the paper and will be of interest to others in the field. I suggest the authors describe their method in much more detail. As presented currently, it is not possible to reproduce it.

We used “On-the-fly MD” achieved by VASP to accelerate the exploration of the potential energy surface during the construction of the training set, primarily for constructing the initial training set. This adaptive methodology dynamically trains machine-learned force field during MD simulations. At each step, machine-learned force field predicts the energy and forces and determines whether the uncertainty exceeds a threshold through Bayesian error estimation. If it does, an initial DFT calculation is performed, and the resulting configuration is added to the training set to update the machine-learned force field. If it is exceeded, an initial DFT calculation is performed and the resulting configuration is added to the training set, updating the machine-

learned force field. This iterative process ensures that the machine-learned force field learns the potential energy surface in real time, focusing computational effort on poorly predicted regions.

In our study, the “Monte Carlo (MC) method” is implemented through our self-developed procedure, which employs a random atom-swapping strategy to rapidly perturb the system. For example, we randomly swap metal atoms in the upper and lower layers of the SMMS, thus accelerating the exploration of the potential energy surface.

The dataset has been uploaded to the Zenodo platform (<https://doi.org/10.5281/zenodo.13777360>).

We have provided a more comprehensive description of the training process in the Supplementary Information (page 1):

“In this study, we adopted a basic iterative training procedure that begins with generating an initial dataset from quantum mechanical calculations, which is then used to train a machine learning model. Next, we validate the reliability of the model and apply it to problems under investigation. If it becomes problematic, e.g., unstable molecular dynamics and poor phonon spectral correspondence, fine-tune it by adding relevant DFT constitutive data. This data expansion and model retraining process are repeated iteratively until the model achieves satisfactory accuracy. The final model can then be deployed for efficient and accurate large-scale atomistic simulations. Then, we employed on-the-fly molecular dynamics (MD) simulations to generate the initial training set, effectively exploring unknown configuration spaces while minimizing sampling costs. Using VASP's on-the-fly MD framework, we implemented the following parameters to optimize efficiency: a timestep of 3.5 fs ensured accurate atomic motion sampling, 20,000 maximized simulation steps, fixed unit-cell shapes, Nosé-Hoover thermostats, and a temperature range (500 K to 1300 K) covered typical growth conditions. The ML_MB=2000 parameter constrained the number of local reference configurations in the on-the-fly MLP. Initial configurations comprised monolayer MoS₂/WS₂ systems and their combinations with homogeneous/heterogeneous metal clusters.

We expanded the training set through ten iterative epochs with distinct sampling strategies:

- **Epoch 1:** On-the-fly MD sampling of MSMS structure dynamics
- **Epoch 2:** Monte Carlo (MC) sampling of MSMS-to-SMMS transitions via random metal-sulfur exchange
- **Epoch 3:** MC sampling of alloyed SMMS intermediates through metal atom swapping
- **Epoch 4:** MC sampling of a sulfur layer on SMMS via metal-S surface atom exchange
- **Epoch 5:** Perturbed configurations near SMMS equilibrium states
- **Epoch 6:** On-the-fly MD sampling of sulfur molecular dynamics (matching initial protocol)
- **Epoch 7:** S-S/Mo-S/Mo-W dimer configurations at varied distances

- **Epochs 8–17:** MLPMD simulations of SMMS growth and Mo deposition (key structures extracted every 10,000 steps)

Iterative model refinement incorporated both historical and new data. We randomly sampled 50,000 data points from newly generated configurations (with some structural repetitions being inevitable during sampling), which we defined as the new data. During each epoch, we constructed a training set of 100,000 configurations through 1:1 random sampling of historical data (cumulative from prior epochs) and newly generated data. The MLP fine-tuning employed a learning rate of 1×10^{-4} to balance convergence speed and accuracy. The final dataset contained approximately 26,000 configurations from initial training and all epochs, ensuring structural diversity and representativeness critical for robust MLP development. The dataset has been uploaded to the Zenodo repository (<https://doi.org/10.5281/zenodo.13777360>)”

Comment 2: *One of the central claims of the paper, appearing in the last sentence of the abstract, is that the SMMS phase that they identify is an “ideal electrode for MoS₂ FETs”. This claim is weakly supported. The two pieces of evidence for the claim are 1) SMMS is a metal, as evidenced by the band structure in Fig. 5a-b, and 2) the p-type Schottky barriers seem to be in a reasonable range. To evaluate the claim that this structure is an “ideal electrode”, much more information would need to be presented, including comparison with existing electrodes, assessment of how easy SMMS is to synthesize relative to these other electrodes, etc. The authors also seem to suggest that the difference between 0.55 and 0.69 eV (the Schottky barriers for SMoMoS and SMMS respectively) is significant, but this seems hard to believe without additional context.*

Our reply: We sincerely appreciate the reviewer's insightful comments regarding our discussion of SMMS as a potential electrode material. We have revised the relevant sections to provide better context and address these concerns:

1. In-situ Electrode Formation: SMMS offers the unique advantage of enabling electrode formation within the MoS₂ layer itself, effectively converting part of the MoS₂ into an electrode.

2. Tunable Schottky Barrier: The difference in Schottky barriers between SMoMoS (0.55 eV) and SMMS (0.69 eV) demonstrates the potential for barrier height control through alloy composition.

3. Reduced Fermi Level Pinning: The SMMS-MoS₂ interface exhibits weak Fermi level pinning ($\Delta\text{SBH} \approx 0.1$ eV) [Fig. R11], in contrast to conventional metal electrodes which show strong n-type SBH and consequently high p-type SBH (1.56-1.75 eV) [ACS Nano 11, 1588–1596 (2017)].

To improve the precision and scientific accuracy of our discussion, we have replaced the term "ideal electrode" with "promising candidate electrode."

Fig. R9: (a-d) Interfacial electronic properties of TMD heterostructures calculated using PBE functional. (a, b) LDOS analysis at the interfaces: (a) SMoMoS-MoS₂ interface (SBH \approx 0.5 eV) and (b) alloyed SMMS-MoS₂ interface (SBH \approx 0.6 eV). Upper panels display atomic configurations (S: yellow, Mo: green, W: white). Lower panels show LDOS comparison between intermediate structures (SMoMoS/SMMS: blue dashed lines) and MoS₂ (green solid lines), with Fermi level aligned at 0 eV. (c, d) Schottky-Mott limit analysis for p-type contacts: (c) SMoMoS-MoS₂ interface exhibits 0.39 eV SBH, (d) SMMS-MoS₂ interface exhibits 0.45 eV SBH.

We have this description in revised main text (page 15):

“

As previously discussed, the SMMS intermediate structure impedes the growth of non-alloyed MoS₂/WS₂ vdWHs. As shown in Fig. 5(a) and Fig. 5(b), the electronic band structure shows that both SMoMoS and SMMS exhibit metallic conducting characteristics. We further investigate the contact characteristics of these metallic intermediate structures with the semiconductor MoS₂. In both cases, the contact between the metallic intermediate structures and the semiconductor MoS₂ forms a p-type Schottky contact, as shown in Figs. 5(c) and 5(d). The Schottky-Mott limit predicts p-type SBHs of 0.55 eV for SMoMoS and 0.69 eV for alloyed SMMS. For conventional metal electrodes (e.g., Ti, Cr, Au, Pd) interfaced with MoS₂, the predicted SBH ranges from 0.56 eV to 1.86 eV⁴². However, Fermi-level pinning induces strong n-type behavior at MoS₂ interfaces. Experimentally observed p-type SBHs range from 1.56 to 1.75 eV. We constructed SMMS-MoS₂ and SMoMoS-MoS₂ interfaces. Compared to Schottky-Mott predictions, we observed

significantly reduced Fermi-level pinning (Fig. S15). The *p*-type SBH increased by only approximately 0.1 eV in the same theoretical framework. Moreover, continuous SBH tuning can be achieved by adjusting metal deposition parameters, enabling tailored contact properties for electronic or optoelectronic applications. These results suggest that SMMS and SMOmOS are promising candidates for MoS₂ FET electrodes.

”

This reference has been added in the revised manuscript:

“

42. Kim, C. *et al.* Fermi Level Pinning at Electrical Metal Contacts of Monolayer Molybdenum Dichalcogenides. *ACS Nano* **11**, 1588–1596 (2017).

”

Comment 3:

Minor concerns

** The authors claim in the SI that the accuracy of their potential is “outstanding”, but the RMSEs don’t strike me as particularly remarkable (151 meV/Å on forces and 10.6 meV on energies). They also claim that their potential “outperforms several existing machine learning force fields in key aspects”, but do not elaborate. For these claims to stand the authors will need to provide direct comparisons with competing potentials. Otherwise, the claims should be removed or significantly weakened.*

Our reply: We appreciate the reviewer's valuable suggestion regarding our description of the machine learning potential's accuracy. The distinctive feature of our potential lies in its ability to accurately describe the complex chemical bond recombination and structural evolution during TMD heterostructure growth. This potential effectively captures the key intermediate structures during the growth process and reveals their crucial role in heterostructure formation. We have thoroughly revised our statements throughout the manuscript to remove subjective characterizations such as "outstanding" and replaced them with more objective, quantitative descriptions.

Comment 4: *It seems the authors do not train on the stress tensor, which is unusual. Why were stress labels excluded from the training set?*

Our reply: We sincerely thank the reviewer for raising this important question. The decision to exclude the stress tensor from our training set was primarily based on the specific application scenario and optimization of our training strategy. Since our molecular dynamics simulations employ the NVT ensemble and focus on atomic-scale structural evolution, the constant system volume renders the direct influence of the stress tensor relatively limited. Under these conditions, ensuring accurate prediction of interatomic forces and system energy has been our primary objective, which has been adequately validated by our current training strategy. We note that in several application cases of the NequIP framework (e.g., *ACS Nano* **2024** *18* (14), 10133-10141), training strategies based solely on forces and energy have proven effective in capturing key dynamic behaviors. This selective training approach not only ensures computational efficiency but

also meets the specific research requirements. We acknowledge the importance of stress tensor training for studies involving lattice deformation or mechanical properties, which will be an important direction for future research expansion.

Comment 5: *In the results section, the authors claim that “the phonon dispersion of the SMOoS structure showed no imaginary frequencies, further confirming that MoSMoS would spontaneously transform into SMOoS.” Lack of imaginary frequencies only confirms that SMOoS is stable, not that MoSMoS would transform into it.*

Our reply: We sincerely appreciate the reviewer's thorough critique. We acknowledge that the original description regarding the formation mechanism of the SMOoS structure requires clarification. Crucially, the MoSMoS configuration was never observed as a transition state in our simulations. Formation energy calculations further indicate its high-energy state, confirming a negligible likelihood of existing as a metastable phase. Therefore, there is a strong driving force for the formation of SMOoS from MoSMoS. In the revised manuscript, we have removed the sentence “the phonon dispersion of the SMOoS structure showed no imaginary frequencies, further confirming that MoSMoS would spontaneously transform into SMOoS.”

Comment 6: *In the results section, when comparing to AIMD simulations, the authors claim that the time for Mo to embed below the surface is significantly shorter than in MLMD simulations “due to the high chemical reactivity of the Mo atoms from vacuum”. I don't understand this explanation. Why isn't this effect captured in the MLMD simulations, which are supposed to reproduce AIMD results?*

Our reply: We sincerely thank the reviewer for pointing out this critical issue. The actual difference stems from the initial condition settings of the two simulations. In the comparative case, the AIMD simulation directly adopts the initial configuration of vacuum deposition, while in the MLPMD case, the Mo atoms are pre-adsorbed on the surface and undergo structural optimization. This difference in the initial configuration's potential energy surface leads to distinct dynamic behaviors. It is important to emphasize that when identical dynamic deposition conditions are applied, the MLPMD also exhibits rapid embedding. This phenomenon validates the MLP's ability to accurately capture atomic-level dynamics. In the revised manuscript, we will clearly distinguish between the two simulation scenarios: the relaxation process of pre-adsorbed structures (on the order of 50 ps) and the process of deposition from vacuum (on the sub-ps timescale). We will also remove the inaccurate "high chemical reactivity" explanation to more rigorously reflect the influence of simulation conditions on the dynamic process.

We have this description in the revised main text (page 8),

“Furthermore, we used DFT to validate the key processes described above. Fig. S7(a) illustrates the process of a single Mo atom embedding into the MoS₂ layer. Despite the relatively low initial kinetic energy, the Mo atom quickly embedded into the MoS₂ layer within approximately 700 fs, as indicated by the sudden drop in energy. Notably, compared to pre-adsorbed Mo atoms, vacuum-deposited Mo atoms significantly shorten the embedding time to about 700 fs — much shorter than the approximately 50 ps required for surface-adsorbed atoms in previous simulations.”

Comment 7: *The convex hull presented in Fig. 2j is unclear. The caption and main text highlight “green dots” that don’t appear in the figure. Four points are labelled with letters, but the letters aren’t referred to anywhere. And it’s hard to tell which point corresponds to S₁MoS.*

Our reply: We sincerely thank the reviewer for his/her careful observation. In the revised version, we redrawn the convex hull plot to highlight the green dots [Fig. R13].

Fig. R10: Revised Fig. 2(j).

Comment 8: *The authors claim that “the Mo-S₁ structure always floats on the surface without sinking”. But the simulation is quite short, only 1.4 ns according to Fig. 4. The authors should perhaps claim instead that “the Mo-S₁ structure remains on the surface without sinking for the duration of the 1.4 ns simulation”.*

Our reply: We thank the reviewer for the valuable suggestions. The use of "always" in the original text indeed conveyed an overly absolute tone. We have revised it to "the Mo-S₁ structure remains on the surface without sinking during the 1.4 ns simulation period" to more accurately reflect the observed time frame. This revised phrasing maintains scientific precision while avoiding overgeneralization.

We have this description in the revised main text (page 14)

“In contrast, once a Mo atom bonds with an S atom, the Mo-S₁ structure remains on the surface without sinking for the duration of 1.4 ns [Fig. 4(k)-4(n)].”

Comment 9: *The authors mention that “the 1T phase present in the early stages mostly disappears, indicating that MoS₂ transitions from the 1T phase to the more stable 1H phase as growth proceeds.” It’s very difficult to see this in Fig. S8. The authors should consider pointing out where the 1T phase appears in the figure, and perhaps give schematic examples of the pristine 1H and 1T phases.*

Our reply: We sincerely thank the reviewer for constructive feedback. To better illustrate this important phase transition, we have enhanced the Supplementary Information as follows: We used arrows of different colors to indicate the locations of the 1T and 1H phases in Fig. R14. Additionally, we included atomic structure schematics of both the 1T and 1H phases, highlighting

their key differences in coordination and atomic arrangement. These improvements help readers more intuitively understand the structural characteristics of the two phases.

Fig. R11: (a-b) Schematic diagrams of (a) 1H-MoS₂ and (b) 1T-MoS₂ structures. MD simulations illustrating the growth processes of the second MoS₂ layer on different substrates: (c)-(g) MoS₂ monolayer substrate and (h)-(l) WS₂ monolayer substrate. The newly formed 2H-phase and 1T-phase domains during growth are indicated by orange and purple arrows, respectively.

Comment 10: *There are grammatical mistakes throughout the paper. For example, in the title, "transition metal dichalcogenides heterostructures" should be "transition metal dichalcogenide heterostructures". In the abstract, "a highly stable intermediate easily introduces" should be "a highly stable intermediate that easily introduces", and "Eliminating the alloying contamination... is avoiding SMMS structure" should be "Eliminating the alloying contamination involves avoiding the SMMS structure".*

Our reply: We sincerely thank the reviewer for the meticulous grammatical review. We corrected these grammatical errors following the reviewer's suggestions.

List of Changes

1. On Page 1 of the Supplementary Information, we have added detailed explanations of the iterative training procedure for machine learning potential. This is in response to the first comments from Reviewer 1 and Reviewer 4.

2. On Page 5 of the revised manuscript and Page 3 of the Supplementary Information, we have added a detailed comparative analysis of the relative stabilities of critical configurations (SMMS and MS₂) obtained from MLP and DFT, along with Table S1. This is in response to the second comment from Reviewer 1.

3. On Page 18, we have added a detailed description of the non-equilibrium MD simulations. On Page 7, we have included additional discussions and Fig. S4 to demonstrate that the embedding behavior of Mo atoms is not sensitive to kinetic energy. On Pages 7, 9, 12, and 13, we have added quantitative analyses of the evolution of metal-metal and metal-sulfur bond counts (Fig. S3, Fig. S8, and Fig. S12) to systematically reveal the metal atoms embedding process and sulfurization reaction kinetics. This is in response to the third comment from Reviewer 1.

4. On Page 7, we clarified the origin of structures in Fig. 2(f-i) and revised the discussion to better explain the role of MLPMD and AIMD simulations. This is in response to the fifth comment from Reviewer 1.

5. On Page 10, we have added detailed definitions of N, T, E_{SMoWS}, E, and x in the equations for ΔF and S, clarifying their physical meanings. This is in response to the ninth comment from Reviewer 1.

6. On Page 18, we have added system size details in the "MD" section of the Methods. This is in response to the thirteenth comment from Reviewer 1.

7. On Page 10, we have clarified the origin of the structures in Fig. 3(g-i). This is in response to the fifteenth comment from Reviewer 1.

8. On Page 18, we have added a detailed description of the non-equilibrium MD simulations. This is in response to the second comment from Reviewer 2.

9. On Page 11, we have added detailed discussions and simulation results to address the discrepancy between the thermodynamically stable SMMS structure and the non-alloyed vdWHs observed in experiments. This is in response to the third comment from Reviewer 2.

10. On Page 13, we have included additional MLPMD simulation results (Fig. S10 and Fig. S11(f-j)) and discussions to explain the atomic-level mechanism of non-alloyed MoS₂/WS₂ vdWHs formation during the sulfurization process. This is in response to the third comment from Reviewer 2.

11. On Page 15, we have revised the discussion on SMMS as a potential electrode material, replacing "ideal electrode" with "promising candidate electrode" and adding context on its advantages. This is in response to the second comment from Reviewer 4.

12. On Page 8, we have clarified the difference in embedding times between AIMD and MLPMD simulations by distinguishing the initial conditions. This is in response to the fourth comment from Reviewer 4.

13. On Page 14, we have revised the phrasing to state that "the Mo-S1 structure remains on the surface without sinking during the 1.4 ns simulation period" to ensure scientific accuracy. This is in response to the eighth comment from Reviewer 4.

Response to Reviewer 1

Comment: In this revision, Zhao and coworkers largely addressed inquiries and comments, as well as attempted to make appropriate changes to the manuscript. While many new details were conveyed in the revision, raised concerns about reproducibility, quality of the MLP, quality of the simulations were not satisfactorily rectified. Furthermore, with the new understanding, new questions emerged about the validity of the simulations in relation to the experimental insights (Nature 621, 499 15 (2023)), they are investigating. These concerns are paired with a remaining lack of readability/clarity of the manuscript, that despite being commented by other reviewers, has not been properly addressed. It would be advisable to put more care into the written text and scientific reporting which still appears to convey numerous mistakes. Compared to the first version, where the benefit of doubt about unknowns prevailed, their work appears now less rigor, raising doubt whether it is publishable.

Author reply: We thank the reviewer for careful reading of our revised manuscript and response to reviewers' comments. We also thank the reviewer for raising more questions about our revised manuscript, which helped us to improve the research greatly. After further study and careful revision of the manuscript, we hope the reviewer finds the new version of the manuscript acceptable. In the following part, we answered the questions and comments by the reviewer point-by-point.

Comment 1: Details on MLP training procedure which the authors added are still vague. For the final data set, the authors should at least explain the model size (e.g. number of atoms & unit cells) and the number of configurations in the training and testing set (to assess the actual number of data-points = energies & forces). The manuscript should be self-contained and it can't be expected

of the reader to download the **Zenodo repo** to answer these questions. An exact understanding of the training set is critical, since by now it becomes clear that the training process was manually and not systematically performed.

Author reply: We thank the reviewer for the suggestion on improving the clarity of our machine-learning potential training dataset description. We have carefully retrained our machine-learning potentials by including more systematic data. The corresponding **SI-data.extxyz files** of our dataset are provided. The dataset comprises 1.18×10^7 data points (26,434 energy points and 1.18×10^7 force points (3.9×10^6 atoms)), covering all key structures encountered during transition-metal dichalcogenide growth. Detailed statistics of our dataset are shown in **Supplementary Table S2** and **SI-data.extxyz files**.

In the revised manuscript, we have added the following table, on Supplementary Information page 6:

Table S2: Dataset statistics (including cell volumes)

Structure Category	Configurations	Avg. Atom Count	Typical Cell Size (Å)
TMD surface with W-Mo-S cluster interaction	18396	155	$22 \times 22 \times 30$
Single-layer Mo on WS ₂ surface	668	196	$22 \times 22 \times 21$
Disordered alloyed TMD structures	1,231	144	$22 \times 22 \times 23$

Structure Category	Configurations	Avg. Atom Count	Typical Cell Size (Å)
1T-phase TMD	1804	144	$22 \times 22 \times 23$
Further Sulfidized Structures of SMMS	360	280	$21 \times 21 \times 21$
TMD nanotubes	403	62	$32 \times 32 \times 5$
Disordered SMMS structures	900	196	$22 \times 22 \times 21$
Ordered SMMS structures	748	144	$18 \times 16 \times 30$
MoSWS structures	96	143	$19 \times 16 \times 22$
Pristine mono-/bilayer MoS ₂ /WS ₂ and heterostructures	138	58	$13 \times 10 \times 26$
Elemental sulfur	1101	35	$14 \times 14 \times 19$
Dimers	589	2	$25 \times 25 \times 25$
Total	26434		

Comment 2: The various methods to create training configurations need much more detailed explanation. For example, using VASP's MLP-MD feature to get data points raises the questions,

whether the data that went to training the Nequip potential were DFT calculations or VASP's MLP prediction (which would introduce considerable noise). Also, the explanation of the Monte Carlo method implemented by the authors remains unclear and should be described in more detail. Quite confusing is the also the wording and usage of words like "epoch" that appears out-of-place and mixes with terminology of NN training (e.g. as used in Nequip). The process of drawing 100k samples from a much larger structure pool, leading to excessive redundancy of 74% also raises questions of good-practice.

Author reply: We thank the reviewer for the insightful comments on our training-set construction. We have revised the manuscript to provide a more detailed descriptions of how the training configurations were created. We address the technical details point by point as follows:

1. Use of the VASP on-the-fly MD feature

We confirm that all data used to train the Nequip potential originate from full self-consistent DFT (SCF) calculations; no interpolated data predicted by any ML potential were included. In the VASP on-the-fly MD framework, if the predicted uncertainty exceeds the threshold value (ML_CTIFOR), a full DFT SCF calculation is automatically triggered. The ionic steps containing electronic self-consistency information are then extracted from vasprun.xml and added to the training set.

2. Monte Carlo sampling method

We agree that the previous explanation of MC sampling method was insufficient and have expanded in the revised manuscript. Our method involves random atomic exchanges between Mo and W atoms across different layers, starting from a relaxed S-Mo-W-S configuration. Each exchange generates a new alloyed structure, and all generated configurations are accepted. While

this approach does not follow the Metropolis algorithm, it is commonly referred to as a generalized Monte Carlo method in materials modelling, where the goal is to efficiently explore configuration space and enhance structural diversity. To avoid confusion, we now describe it as a “randomized atom-exchange sampling scheme” in the revised manuscript.

3. Terminology normalization

We appreciate the reviewer’s comment regarding the confusion caused by the term “epoch”. To reduce ambiguity, all occurrences of “epoch” have been replaced by “training round” throughout the manuscript.

4. Clarification of the 100k sampling strategy

The alleged “74% redundancy” does not occur in our workflow. In each training round, we randomly draw historical and newly generated data with a 1:1 ratio to fine-tune the MLP model. This balance prevents catastrophic forgetting while continuously introducing new chemical environments. The use of “100k” is simply a practical implementation detail of this sampling strategy. To avoid misunderstanding, we have revised the manuscript to clarify this point.

We also modified this in Supplementary Information on page 1:

“In this study, we adopted a basic iterative training procedure of MLP that begins with generating an initial dataset from quantum mechanical simulations. Next, we validate the reliability of the model by comparing the data with DFT and try to apply it to simulate the 2-step growth of MoS₂. If the MLP results in unstable molecular dynamics and poor phonon spectral correspondence, we then fine-tune the MLP by adding more relevant DFT constitutive data. These data expansions and models retraining processes are repeated iteratively until the model achieves

satisfactory accuracy. The final model can then be deployed for efficient and accurate large-scale atomistic simulations.

Then, we employed on-the-fly molecular dynamics (MD) simulations to generate the initial training set, effectively exploring unknown configuration spaces while minimizing sampling costs. Using VASP's on-the-fly MD framework, we implemented the following parameters to optimize efficiency: a timestep of 3.5 fs ensured accurate atomic motion sampling, 20,000 maximized simulation steps, fixed unit-cell shapes, Nosé-Hoover thermostats, and a temperature range (500 K to 1300 K) that covered typical growth conditions. The $ML_MB=2000$ parameter constrained the number of local reference configurations in the on-the-fly MLP. Initial configurations comprised monolayer MoS_2 and WS_2 and their combinations with homogeneous/heterogeneous junctions and adsorbed metal clusters. We expanded the training set through iterative training rounds with distinct sampling strategies:

- Training round 1: On-the-fly MD sampling of MSMS structure dynamics
- Training round 2: Randomized metal-sulfur exchange sampling — Starting from the relaxed MSMS structure, we performed random substitutions between metal (Mo/W) and sulfur atoms at specific interfacial sites to generate intermediate SMMS-like configurations. These exchanges emulate early-stage metal–chalcogen mixing observed during the deposition process.
- Training round 3: Randomized metal-metal swapping — To construct a diverse set of alloyed SMMS structures, we performed layer-constrained Mo–W swaps across top and bottom layers. This preserves the overall composition but introduces configurational disorder relevant to alloying behaviour.

- Training round 4: Surface metal–sulfur exchange sampling — Starting from SMMS structures, we randomly exchanged surface S atoms with underlying metal atoms (Mo or W) to simulate possible sulfurization pathways and local rearrangements during the growth process. atom exchange

- Training round 5: Perturbed configurations near SMMS equilibrium states

- Training round 6: On-the-fly MD sampling of sulfur molecular dynamics (matching initial protocol)

- Training round 7: S-S/Mo-S/Mo-W dimer configurations at varied distances

- Training rounds 8-17: MLP-MD simulations of SMMS growth and Mo deposition (key structures extracted every 10,000 steps).

During each Training round, we fine-tune the MLP model through 1:1 random sampling of historical data (cumulative from prior training rounds) and newly generated data. The MLP fine-tuning employed a learning rate of 1×10^{-4} to balance convergence speed and accuracy. The final dataset contained approximately 26,000 configurations from initial training and all training rounds, ensuring structural diversity and representativeness critical for robust MLP development. The dataset has been uploaded to the **Zenodo repository** (<https://doi.org/10.5281/zenodo.13777360>) and included in **SI-data.extxyz**”

Comment 3: The MLP validation still falls short. From the revision it becomes apparent that no uncertainty metric was used during training (e.g. to track convergence), and instead qualitative criteria “unstable molecular dynamics” or “poor phonon spectra” called for “refining”. This should be clearly labelled as an ad-hoc approach and validation of the MLP needs to be especially meticulous. The comparison of “critical structures” is a good first step, but by far not enough to determine the MLPs accuracy over the investigated composition and structure space. A sign of significant discrepancy between MLIP and DFT is Fig. 3(j), and certainly raises doubt over the statement on page 8 “accurately reproduced DFT results”.

Author reply: We thank the reviewer for raising this critical point regarding the MLP validation. The configuration shown in Fig. 3(j) is thermodynamically unstable due to its relatively high energy, leading to insufficient accuracy of the MLP force constants for this structure. **We have carried out targeted re-sampling of this configuration and incorporated perturbed variants into the training set. Specifically, we added 668 configurations** obtained from VASP on-the-fly AIMD sampling of a single-layer Mo on WS₂ surface, as well as 96 perturbed configurations of the MoSWS structure, to enrich the relevant local configuration space. After retraining the MLP with the expanded dataset, the agreement between the MLP and DFT phonon spectra has improved substantially. Although minor frequency deviations remain, the overall dispersion now reproduces the DFT result reasonably well (see new Fig. R1).

We also replaced the Fig. 3(j) accordingly.

Figure R1: Phonon dispersion of the MoSWS.

Comment 4: The authors did not implement the previous inquiry about systematic simulations and statistics. While a more thorough data analysis is appreciated, also more test simulations (starting from different initial conditions) should be performed – which is easily achievable with the highly performant Nequip potential. But also, in the newly provided analysis, many questions arise: Why do the number of bonds start at zero in Fig. S3 and S8? What about existing bonds? Those seem to be accounted for in Fig. S12. Note also, S3 and S8 have the same caption, which is probably another mistake. Further, it might make sense to distinguish the bonds between different elements, which could help understand the growth process.

Author reply: We thank the reviewer for further raising the issue of testing simulations. **We have performed more simulations to verify the methodology and the conclusions presented in the manuscript.**

1. More test simulations from multiple initial conditions: We thank the reviewer for the suggestion regarding statistical repetition. After careful consideration, we believe that, for the embedding process investigated in this work, performing multiple simulations from randomized initial velocity distributions is not scientifically necessary. First, from a

theoretical standpoint, DFT energy calculations indicate that the embedding of an Mo atom in MoS₂ is strongly thermodynamically favoured, with an energy release of ~1.45 eV. Second, in the present study, we have repeatedly observed consistent embedding behaviour across a variety of independent simulations, including different substrates (MoS₂ and WS₂), temperatures (300–900 K), and incident kinetic energies. While the initial velocities in these simulations were not explicitly constructed for statistical sampling, they effectively serve as physically distinct perturbations, under which the same qualitative behaviour was observed. This indicates that the embedding process is robust and not sensitive to the specific initial conditions. Based on these considerations, we believe that additional randomized-velocity simulations, while potentially helpful, would not change the physical conclusions we draw regarding the embedding mechanism. We nonetheless acknowledge the general value of such statistical tests and plan to explore them further in future studies involving more complex dynamical pathways.

2. Clarification on Figs. S3, S8, and S12: Regarding the issue of zero points in bond counts, we provide a complete physical explanation: the adopted cutoff radius of 2.85 Å is larger than typical M-M bonds in M bulk (2.74 Å and 2.75 Å for Mo and W, respectively) but smaller than the metal-metal equilibrium spacing in MoS₂/WS₂ (3.15 Å). The essential difference between Fig. S12 and S3/S8 lies in the different observation objects: S12 records the change in Mo/W-S bond counts during the sulfurization process, and its initial value >0 reflects the existence of inherent metal-sulfur bonds in the substrate. While S3/S8 records the number of newly formed metal bonds during the deposition process, which does not include the initial bonding state of the substrate.

3. Distinguishing Bonds Between Different Elements: We thank the reviewer for suggesting the differentiation of metal–metal bonds (Mo–Mo, Mo–W, and W–W). We have added Fig. R2, which presents the time evolution of different types of metal–metal bonds during the deposition of Mo atoms onto a WS₂ substrate. A cutoff distance of 2.85 Å was used for bond identification. As shown in Fig. R2, the number of Mo–Mo and Mo–W bonds increase rapidly during the early stages of deposition, reflecting the aggregation of Mo atoms and their bonding interactions with the underlying W atoms. In contrast, the number of W–W bonds increase more slowly. This increase does not represent the formation of new chemical bonds but rather results from local lattice rearrangements during the deposition process. Specifically, some W–W pairs that originally had interatomic distances greater than the cutoff (the W–W distance in pristine WS₂ is approximately 3.15 Å) experience slight displacements or strain-induced contractions, causing their separation to fall below the 2.85 Å threshold and thus be counted as bonds in the analysis. Therefore, the gradual rise in W–W bond count should be interpreted as a reflection of subtle structural reconstructions in the static lattice induced by Mo atom deposition, rather than the creation of new W–W bonds.

Figure R2: Number of Mo-Mo, Mo-W, and W-W bonds during Mo deposition on a WS₂ substrate. A bond cut-off distance of 2.85 Å was applied for bond identification in both cases.

In the revised manuscript, we have modified these captions in Supplementary Information, on page 8:

“**FIG. S3:** Number of Mo-Mo bonds during Mo deposition on a MoS₂ substrate. Bond identification is based on a cut-off distance of 2.85 Å, which is larger than the Mo-Mo bond length in Mo bulk (~2.74 Å) but smaller than the equilibrium Mo-Mo spacing in pristine MoS₂ (~3.15 Å).”

On page 12:

“**FIG. S8:** (a) Evolution of the total number of Mo/W-Mo/W chemical bonds (including Mo-Mo, Mo-W, and W-W bonds) during Mo deposition on a WS₂ substrate. (b) Individual counts of Mo-Mo, Mo-W, and W-W bonds. Bond identification is based on a cut-off distance of 2.85 Å, ,

which is larger than the Mo-Mo bond length in Mo bulk (~ 2.74 Å) and W bulk (~ 2.75 Å) but smaller than the equilibrium Mo–Mo spacing in pristine MoS₂ (~ 3.15 Å)”

On page 15:

“**FIG. S12:** (a) Evolution of the number of Mo-S bonds during sulfur deposition on the SMoMoS intermediate structure. Excess sulfur atoms extract Mo atoms from the SMoMoS structure, leading to the formation of a bilayer MoS₂ heterostructure. (b) Evolution of the number of Mo/W-S bonds during sulfur deposition on the alloyed S(Mo_{0.5}W_{0.5})(W_{0.5}Mo_{0.5})S structure. Sulfur atoms facilitate simultaneous extraction of both Mo and W atoms, ultimately forming an alloyed bilayer Mo_{0.5}W_{0.5}S₂ configuration. A cut-off distance of 2.6 Å was applied for both Mo-S and W-S bond identifications. The bond counts include both pre-existing and newly formed bonds in the system. The initial number of bonds reflects the presence of metal–sulfur coordination in the intermediate structures, and as the bilayer TMD forms, the number of Mo–S bonds nearly double—indicating the completion of coordination environments in the emerging bilayer structure.”

Comment 5: The revised description of the simulations still requires careful refinement. It is not, yet, fully clear what is done and in the current description, terminology of ensembles is mixed up.

Author reply: We thank the reviewer for pointing out the ambiguity in our description of the simulation ensembles. After careful review, we confirm that, except for the AIMD validation of Mo-atom embedding shown in Supplementary Figure S7(a), which employs the NVE ensemble, all other MLP-MD and AIMD simulations were carried out in the canonical (Nosé–Hoover NVT)

ensemble. To avoid any further confusion, we revised our manuscript to state clearly the ensemble and temperature for each simulation.

In the revised manuscript, we have added the following Table, on Supplementary Information page 7:

“Table S3: Summary of simulation parameters used in each figure

Figure Reference	Temperature	Ensemble
Fig. 2(d-e)	900 K	NVT
Fig. 3(a-f)	900 K	NVT
Fig. 4	1100 K	NVT
Fig. S4	900 K	NVT
Fig. S6	1100 K	NVT
Fig. S7(a)	1100 K	NVE
Fig. S7(b-c)	1100 K	NVT
Fig. S9	1100 K	NVT
Fig. S10	300 K	NVT
Fig. S11	1100 K	NVT
Fig. S13	1100 K	NVT
Fig. S14	1100 K	NVT
Fig. S15	900 K	NVT
Fig. S16	900 K	NVT
Fig. S17	1100 K	NVT
Fig. S18	1100 K	NVT

”

Comment 6: The discussion of the manuscript results still requires better structure, a lot of back-and-forth is contained around Fig. 3 making it hard to follow the authors argumentation. After revision, new questions about the simulations occurred. For example, a key result is the fact, that “metal films do not float” (compared to the original experimental data in Nature 621, 499 15 (2023)), what is the significance of this finding?

Author reply: Thank the reviewer for concern about our discussion. We have reorganized the description of Figure 3 in the “Results and Discussion” section into four steps—empirical observation, mechanistic discussion, kinetic validation and statistical analysis. Specifically, at 900 K, our MLP-MD simulations show that Mo atoms deposited on the WS₂ surface quickly diffuse through the top S-layer, which contrast the sharply with the “coating” hypothesis of Nature 621, 499 (2023). This indicates that the growth mechanism presented in the previous study is questionable, and thus the route towards the synthesis of wafer-scale vdWHs requires more experimental validation.

In the revision, we expanded discussion on the “metal films do not float” finding to address its significance and possible reasons for divergence from experimental observations. On manuscript page 11:

“A recent study reported²³ that a two-step vapor deposition process was used to synthesize wafer-scale TMD vdWHs, where the key intermediate structure during growth is a Mo atomic monolayer on the WS₂ surface. However, our MLP-MD simulations reveal that the Mo atomic monolayer on the WS₂ surface is highly unstable, it can be easily transformed into the SMOWS structure by thermal annealing. Then the growth kinetics is actually different from that presented in the reference²³.”

Comment 7: The results also contain quite a bit of inconsistent simulations: The MD simulations are conducted around deposition temperatures of 500K-1300K (?), but some analysis (e.g. mixing entropy), and other simulations (e.g. depicted in Fig. R7) are conducted at 300K. This seems inappropriate if the synthesis mechanism is investigated. Some arguments are made about alloying where WS₂ and MoS₂ slabs appear not systematically exchanged and kinetic stabilization is argued on long MD timescales (“stability during 1.4 ns”) but without a significant number of samples. These simulations thus only weakly support the authors’ conclusions.

Author reply: We thank the reviewer for this important and detailed comment. We understand the reviewer’s concerns regarding the use of different temperature settings in our simulations. This issue may stem from the presence of multiple simulation modules in the manuscript, each serving distinct purposes. We clarify the physical background and motivation for the different temperature choices as follows:

The 500–1300 K temperature range was used exclusively in the “on-the-fly” sampling phase of AIMD, aimed at constructing the training dataset for the MLP (as detailed in SI Section 1). These simulations were designed to sample a wide range of atomic configurations that may occur during growth processes and were not intended to reflect the actual physical conditions of deposition.

Simulations conducted at 300 K serve two specific purposes. First, this temperature was used as the reference in our mixing free energy calculations, allowing us to assess the thermodynamic stability of alloyed versus non-alloyed structures under ambient conditions. Second, one MLP-MD simulation at 300 K (Fig. S10) was performed to examine whether Mo–W atomic exchange could

occur at low temperature. The result showed that such exchange is kinetically suppressed at 300 K, which provides a plausible explanation for the formation of non-alloyed interfaces observed in experiments.

Regarding the reviewer's comment that we lack systematic simulations of MoS₂ and WS₂ structure substitutions, we appreciate this insightful suggestion. We have supplemented the revision with simulations of the behaviour of Mo-based clusters on WS₂, and Fig. R3 presents the results under the same simulation conditions (1100 K) as those used for MoS₂.

Furthermore, concerning the reviewer's concerns about the MD simulation time (1.4 ns) and the limited number of samples, we would like to further clarify the objectives and simulation strategy of this work. Our research is not aimed at accurately calculating diffusion coefficients or constructing complete statistical distributions but rather focuses on the differences in the dynamic behaviour of different precursor clusters on the TMD surface, thereby revealing their structure-dependent behaviour and potential reaction pathways.

Despite the limited total simulation time, we still observed clear cluster-dependent dynamic behaviour. As shown in Fig. R3 (e, j, o, t), different types of Mo-based clusters exhibit noticeably different in-plane migration behaviours on the WS₂ surface:

Mo single atom undergoes localized motion and are trapped in the initial region within a short time (Fig. R3e);

Mo–S and Mo–S₂ clusters exhibit limited diffusion on the surface, with lower mobility (Fig. R3j, Fig. R3o);

In contrast, Mo-S₃ clusters exhibit more significant surface mobility and irregular migration paths (Fig. R3t).

To further verify the influence of temperature on dynamic behaviour, we also conducted similar simulations at 900 K (see Fig. R4). The results show that the diffusion behaviour of Mo-S and Mo-S₂ clusters is significantly reduced at the lower temperature, which contrasts with the migration trends under high-temperature conditions.

Figure R3: MLP-MD simulations of various Mo-based clusters on WS₂ (1100 K): (a-d) Single Mo atom deposition; (e) Mo atom trajectory on the xy plane; (f-j) Mo-S cluster; (k-o) Mo-S₂ cluster; (p-t) Mo-S₃ cluster.

Figure R4: MLP-MD simulations of various Mo-based clusters on WS₂ (900 K): (a-d) Single Mo atom deposition; (e) Mo atom trajectory on the xy plane; (f-j) Mo-S cluster; (k-o) Mo-S₂ cluster; (p-t) Mo-S₃ cluster.

In summary, although extending the simulation time or increasing the sampling of initial conditions could further improve quantitative accuracy, in the current setup, we have been able to identify the migration trends and relative activity of different cluster structures on the TMD surface. These results provide meaningful physical insights into the structural evolution and surface reconstruction of metal precursors in the early stages of deposition.

In the revised manuscript, we have added the following sentences, on manuscript page 14:

“To deeply understand the dynamic behaviour of Mo atoms and their clusters with S atoms on various TMD substrates, we conducted further MLP-MD simulations to investigate the migration mechanism and stability of Mo clusters on WS₂ surfaces. We found that their behaviour was similar to that observed on MoS₂ substrate (Fig. S14). Fig. S15 and S16 illustrate the behaviour of single Mo atoms and Mo-S clusters on MoS₂ and WS₂ surfaces at a lower temperature (900 K). The results indicate that, even at reduced temperatures, single Mo atoms still tend to rapidly embed into the substrates. However, the surface mobility of Mo-S and Mo-S₂ clusters is significantly reduced compared to their behaviour at 1100 K. In contrast, Mo-S₃ clusters maintain relatively high surface stability and mobility, further validating the critical role of sulfur-containing clusters in suppressing Mo atom sinking (Fig. S15, Fig. S16). Fig. S17 further confirms the reliability and accuracy of these MLP-MD results through AIMD simulations at 1100 K for 10 ps. Although the simulation duration was shorter due to computational limitations of AIMD, the clusters displayed similar characteristics as observed in the MLP-MD simulations.”

Response to Reviewer 2

Comment 1: Report on the Revised Manuscript by Zhao et al. 1. The language and overall presentation of the manuscript have been significantly improved, making the paper much more readable and understandable. However, the abstract and the first two paragraphs of the introduction still require further polishing for clarity and conciseness.

Author reply: We sincerely thank the reviewer for the encouraging feedback on the revised manuscript and for acknowledging the improvements in language and presentation. Based on the reviewer's suggestion, we have carefully revised the abstract and the first two paragraphs of the introduction to improve clarity, eliminate redundancy, and enhance conciseness.

In the revised manuscript, we have modified this in manuscript, on page 1:

“Abstract

Two-dimensional (2D) transition metal dichalcogenide (TMD) van der Waals heterostructures (vdWHs) hold promise for next-generation electronics, but their large-scale synthesis remains limited by size constraints and alloying contaminations. Recently, a two-step vapor deposition method was reported for growing wafer-size TMD vdWHs with minimal impurities [Nature 621, 499 (2023)]. In this study, we developed a machine learning potential (MLP) that accurately captures the atomic-scale dynamic growth process of bilayer MoS₂/WS₂ vdWHs under feasible growth conditions. Our simulations uncover a crucial metastable SMMS (M = Mo or W) intermediate structure that facilitates metal atom swap and alloying. Eliminating the alloying contamination requires preventing the embedding of bare metal atoms. Interestingly, the SMMS

structure also exhibits favourable electronic properties and emerges as a promising low Schottky barrier contact electrode for MoS₂ field-effect transistors (FETs).

Introduction

2D TMDs have attracted intense attention due to their suitable band gaps and high carrier mobility¹⁻⁴, strong nonlinear optical response^{5,6}, and ease of layer stacking assembly. The integration of vdWHs by pristine TMDs layers can significantly enhance their properties, enabling immense potential applications in microelectronics, optoelectronics, and nonlinear optics⁷⁻¹⁴.

Despite their promise, the controlled growth of TMD vdWHs remains numerous challenges. While commonly used mechanical assembly methods can achieve high-quality TMD vdWHs with atomically sharp interfaces^{15,16}, they often struggle to achieve wafer-size dimensions and can be prohibitively expensive. In contrast, chemical vapor deposition (CVD) has achieved significant success in growing wafer-sized monolayer TMD^{12,17,18}. However, growing TMD vdWHs still faces limitations in size constraints and a strong tendency toward alloy formation^{19,20}. Among these scalable methods, metal-organic chemical vapor deposition (MOCVD) has emerged as a promising alternative for growing large-size TMD heterojunctions with improved interfaces²⁰⁻²².

Comment 2: The authors now clarify that they used an ideal Morse potential for the substrate simulation. Given that the study is motivated by experimental results on sapphire, it is essential that the manuscript discusses how well this idealized simulation approach approximates the actual experimental conditions. A brief justification or discussion in the main text would strengthen the manuscript.

Author reply: We thank the reviewer for this insightful suggestion regarding the rationale of the substrate model. To emphasize the focus of this study on the internal interactions within the Mo–W–S ternary system, we simplified the substrate in the simulations to a single rigid hard wall—constraining the vertical motion of TMDs via an ideal Morse potential without introducing any chemical adsorption or interfacial reactions. Substrates such as sapphire ($\alpha\text{-Al}_2\text{O}_3$) and silica (SiO_2) are commonly used in experiments, and their interactions with transition metal sulfides are typically weak (adsorption energies < 0.1 eV/atom), much lower than the Mo–S or W–S bond energies (>1 eV/bond). Therefore, their primary role in our context is limited to mechanical support rather than chemical bonding. Our simplified model avoids the uncertainty of additional parameters while maintaining an efficient and clear exploration of the embedding mechanism.

In response, we have added the following discussion to the manuscript (Method Section), on page 19:

“This wall mimics the weak van der Waals interaction (< 0.1 eV/atom) between Mo/W atoms and oxide substrates such as sapphire, thus providing only mechanical support while avoiding spurious interfacial chemistry.”

Comment 3: The authors have added two supplementary figures to clarify the heterostructure formation. Figure S10 includes the simulation temperature, but this detail is missing in Figure S11. For clarity, the temperature should also be included in Figure S11 to clearly indicate the conditions under which the sulfurization process was studied.

Once these remaining issues are addressed, I believe the manuscript will be suitable for publication in Nature Communications.

Author reply: We thank the reviewer for pointing out this inconsistency. We agree that including the simulation temperature in Figure S11 is important and beneficial for clarity and reproducibility.

We have revised Figure S11 and added the label “1100 K” to the figure caption.

Response to Reviewer 3

Comment 1: I co-reviewed this manuscript with one of the reviewers who provided the listed reports. This is part of the Nature Communications initiative to facilitate training in peer review and to provide appropriate recognition for Early Career Researchers who co-review manuscripts.

Author reply: We appreciate the transparency regarding the co-review process and fully support Nature Communications' initiative to involve and recognize Early Career Researchers in peer review. We thank both the reviewer and the co-reviewer for their thoughtful and constructive comments, which have greatly helped us improve the quality and clarity of the manuscript.

Response to Reviewer 4

Comment 1: The authors have adequately responded to my comments and I am happy to recommend the manuscript for publication.

Author reply: We greatly thank the reviewer for the recommendation of our paper to **Nature Communications**.

Comment 2: As a small comment, I noticed that in revised Fig. S14, the figure and caption refer to a “2H” phase, but in the rest of the manuscript, this phase is referred to as “1H”. I think the authors probably meant to write “1H” here.

Author reply: We thank the reviewer for the positive and valuable feedback. **We have carefully checked Fig. S14 and corrected the “2H” phase to “1H” phase.**

Comment 3: The code contained in “ocp_active.zip” is unlikely to be a usable resource for the community, as it appears to be available only in the zenodo repository as a set of python files (as opposed to a maintained repository on github), and it is unclear what files from the original Open Catalyst repository were modified or how they were modified. I don’t think this should necessarily preclude publication, however, as the authors describe their methods in the manuscript and do make the code available, and more importantly they also provide their training and test structures in extxyz format, which may very well be useful for the community.

Author reply: We appreciate the reviewer’s concern regarding the accessibility and usability of the code. In response, we have uploaded the implementation to a public GitHub repository: <https://github.com/1713175349/ocp>. This repository includes all modifications made to the original Open Catalyst Project (OCP) framework, and is sufficient to reproduce our training pipeline and results without requiring access to the original OCP codebase.

List of Changes

1. On Pages 1 and 6 of the Supplementary Information, we have added a detailed explanation of the iterative training procedure for our machine learning potential, clarified terminology. We have added Table S2 with comprehensive dataset statistics (configuration counts, average atom numbers, and typical cell sizes), and provided SI-data.extxyz as an attachment. This is in response to comments 1 and 2 from Reviewer 1.
2. On Page 11 of the manuscript, we have reorganized the discussion around Figure 3 and expanded on the significance of our finding that metal atoms embed into the substrate. We have also improved the MLP validation by incorporating 764 new configurations into the training set and presenting an updated phonon dispersion figure (Fig. R1) to demonstrate improved accuracy. This is in response to comments 3 and 6 from Reviewer 1.
3. On Pages 8, 12, and 15 of the Supplementary Information, we have revised the captions for Figures S3, S8, and S12, and included a new differentiated bond analysis (Fig. R2) to systematically clarify the simulation results. On Page 7 of the Supplementary Information, we have added a comprehensive table of simulation parameters (Table S3). This is in response to comments 4 and 5 from Reviewer 1.
4. On Page 14 of the manuscript, we have added discussion and supplementary figures (Figs. S14, S15, S16, S17) presenting new simulations of cluster dynamics on WS₂ at various temperatures to address concerns about simulation consistency and provide further evidence for our conclusions. This is in response to comment 7 from Reviewer 1.
5. On Page 1 of the manuscript, we have revised the Abstract and Introduction for improved clarity and conciseness. On Page 19, we have added a justification for the simplified

substrate model used in our simulations. This is in response to comments 1 and comment 2 from Reviewer 2.

6. In the Supplementary Information, we have added the simulation temperature to the caption of Figure S11 and corrected the phase label in the caption of Figure S14 from “2H” to “1H”. We have also uploaded our complete code to a public GitHub repository to improve accessibility for the community. This is in response to comment 3 from Reviewer 2 and the comments from Reviewer 4.

Response to Reviewer 1

Comment: The authors have implemented many of the requested changes, which has significantly improved the manuscript's transparency and traceability. This removes one of the main concerns raised in the previous revision-rounds. However, I still find that the work falls short in a few critical aspects related to reproducibility, contextualization with experimental findings (especially Ref. [23]), and the overall scientific rigor in reporting. There are also recurring issues in presentation, including inconsistent units and unclear phrasing, which are concerning at this advanced revision stage. Overall, these shortcomings present major revision points and still need careful attention.

Author reply: We sincerely thank the reviewer for the constructive feedback and for acknowledging the significant improvements in the manuscript's transparency and traceability. We have carefully addressed the concerns regarding reproducibility, contextualization with experimental findings, and presentation. Below is our point-by-point response.

Comment 1: MLIP training and validation: The inclusion of the MLIP training and validation details is appreciated and now allows the reader to follow the methodology. The reported RMSEs appear acceptable considering the high-energy structures accessible at elevated temperatures. Nevertheless, to align with state-of-the-art reporting, I suggest the authors should include a measure of convergence — either RMSE versus “training round” or RMSE as a function of structure energy per atom — to assess model quality comprehensively. I also recommend revising the newly added paragraphs for clarity, as the current language is rough (especially the first one). Units for energy RMSE (meV vs. meV/atom) must be reported consistently.

Author reply: We appreciate the reviewer's valuable suggestions regarding the validation of our machine learning potential (MLP). To comprehensively assess the model quality, we adopted the reviewer's recommendation to analyse the RMSE of energy as a function of the formation energy per atom (calculated relative to the elemental references), as shown in Fig. R1. We believe this metric is particularly informative as it demonstrates the model's accuracy across a diverse configuration space, ranging from stable low-energy structures to high-energy non-equilibrium states encountered during growth. We have included this analysis in the Supplementary Information [Figure S1(c)]. The results reveal that the model achieves high accuracy for stable structures and maintains prediction errors within a reliable range (with energy RMSE generally remaining below 20 meV/atom) even for high-energy configurations. Furthermore, we have revised the paragraphs describing the MLP training in the Supplementary Information to improve flow and clarity. We have also carefully checked the entire manuscript and standardized all energy RMSE units to meV/atom and force RMSE units to meV/Å to ensure consistency.

Figure R1: Distribution of energy RMSE for test set structures across varying formation energy intervals. The histograms display the prediction errors for energy in meV/atom. The formation energy per atom determines the binning interval.

On Page 5 of manuscript, we added the related discussions to assess model quality comprehensively.

“To comprehensively assess the model's reliability across the energy landscape, we analysed the prediction errors as a function of structure formation energy [see Fig. S1(c)]. This analysis confirms that the MLP maintains high accuracy not only for stable low-energy configurations but also for high-energy states relevant to growth simulations.”

On Supplementary Information page 4:

“FIG. S1(c) provides a more detailed error analysis by categorizing the test structures based on their formation energy per atom. The histogram displays the energy RMSE distribution across different formation energy intervals. This analysis reveals that the model maintains exceptional accuracy for structures in the lower energy regimes (typically corresponding to equilibrium or near-equilibrium configurations). Crucially, even for higher-energy states, which represent distorted configurations or transition states frequently encountered during high-temperature growth simulations, the energy prediction errors remain low and bounded. This comprehensive validation confirms the MLP's robustness and consistent accuracy across the entire relevant energy landscape.”

On Page 1 of Supplementary Information, we revised the related paragraphs as the reviewer suggested:

“In this study, we employed an iterative training approach to develop the MLP. The process began with the generation of an initial dataset using ab initio molecular dynamics (AIMD) simulations. Subsequently, we assessed the model’s reliability by benchmarking it against DFT calculations and applying it to simulate the two-step growth of MoS₂/WS₂. If the MLP exhibited instabilities during molecular dynamics or failed to reproduce accurate phonon spectra, we refined the model by incorporating additional relevant DFT configurations into the training set. This cycle of data expansion and model retraining was repeated iteratively until the MLP achieved satisfactory accuracy and stability. The final optimized model was then deployed for efficient and accurate large-scale atomistic simulations.

To generate the initial training set, we employed on-the-fly molecular dynamics (MD) simulations, a method that effectively explores unknown configuration spaces while minimizing sampling costs. These simulations were conducted using the VASP on-the-fly MD framework with parameters optimized for efficiency and accuracy. Specifically, we utilized a timestep of 3.5 fs to capture atomic motions precisely, with a maximum of 20,000 steps per trajectory. The simulations were performed in the canonical (NVT) ensemble using Nosé-Hoover thermostats, covering a temperature range from 500 K to 1300 K to encompass typical growth conditions. The unit-cell shapes were kept fixed, and the number of local reference configurations in the on-the-fly MLP was constrained by setting ML_MB=2000. The initial configurations included monolayer MoS₂ and WS₂, as well as their combinations featuring homogeneous/heterogeneous junctions and adsorbed metal clusters. Subsequently, the training set was expanded through iterative training rounds employing distinct sampling strategies.”

Comment 2: Reproducibility: I strongly disagree with the authors’ assertion that performing multiple simulations is unnecessary. Repetition under varying initial conditions is essential, particularly for non-equilibrium simulations, to validate qualitative insights and ensure reproducibility. This is not about obtaining quantitative observables but demonstrating that observed mechanisms are robust. Performing multiple simulations could result in a brief statement such as, “All qualitative observations were confirmed in x independent runs” and would meaningfully strengthen the manuscript. Testing one’s own hypotheses in this way is fundamental to good scientific practice.

Author reply: We acknowledge the reviewer’s concern regarding reproducibility and have performed additional simulations to demonstrate the robustness of our qualitative insights. To address this, we performed three independent MD simulations with different random velocity seeds, focusing on the primary non-equilibrium dynamic processes that underpin our main conclusions. Regarding the deposition process, as shown in Fig. R2, Mo atoms invariably sank into the chalcogen layer within similar timescales across all three independent runs, confirming that the formation of the embedded intermediate is a robust energetic preference. Similarly, for the sulfurization process shown in Fig. R3, all independent runs confirmed that the deposition of excess S atoms successfully pulled metal atoms to the surface, forming the bilayer structure. These

results confirm that our observed mechanisms are highly robust and reproducible. We have added a statement in the "Results and Discussion" and "Methods" sections to attest to this verification.

Furthermore, a very recent study published in *Science* [Gao et al., *Science* 390, 813–818 (2025)] reported an "**atomic layer bonding (ALB)**" contact (Fig. R4). This study realized a direct metallic bonding interface between metal electrodes and the transition-metal layer of TMDs by removing the top sulfur layer. **The physical nature of this ALB structure (direct metal-metal bonding) is highly consistent with our predicted SMMS intermediate.** Crucially, the experimental ALB contact was shown to be metallic and exhibited superb thermomechanical stability and ultralow contact resistance. This experimental evidence strongly supports our theoretical prediction that the SMMS structure serves as a promising electrode candidate. We have incorporated this discussion into the revised manuscript to further substantiate the physical validity and application potential of our identified intermediate.

Figure R2: Snapshots of Mo atom deposition on MoS₂ from 900 K MLP-MD simulations using three different random seeds: (a-d) Run 1, (e-h) Run 2, and (i-l) Run 3.

Figure R3: Snapshots of sulfurization on the alloyed S(Mo_{0.5}W_{0.5})(W_{0.5}Mo_{0.5})S structure from 1100 K MLP-MD simulations using three different random seeds: (a-d) Run 1, (e-h) Run 2, and (i-l) Run 3. The S atoms consistently extract Mo/W atoms to form a bilayer alloyed Mo_{0.5}W_{0.5}S₂ structure.

Figure R4: Comparison of our predicted structure with the experimental ALB contact. (a) The SMoMoS structure identified in our simulations. (b) The ALB structure reported in Gao et al., *Science* (2025). (c) Cross-sectional STEM image of the experimental ALB structure, showing direct metal-metal bonding similar to SMoMoS.

In the revision, we added the following sentence on manuscript page 20:

“To ensure the reproducibility of the non-equilibrium processes described above, all key qualitative observations (including the sinking of Mo atoms and the sulfurization-induced extraction) were confirmed in three independent MD simulations initialized with different random velocity seeds.”

On manuscript page 17:

“Notably, a very recent experimental study⁴³ realized an “atomic layer bonding (ALB)” contact by establishing a metallic coherent bonding interface between the transition-metal layer of TMDs and metal electrodes. This ALB structure shares the key feature of direct metal-metal bonding observed in our predicted SMMS intermediate. Consistent with our findings that SMMS exhibits

metallic character and improved contact properties, the experimental ALB contact demonstrated ultralow contact resistance and superb thermomechanical stability. This strongly corroborates our theoretical prediction that establishing coherent metal-metal interactions (as in SMMS) is an effective strategy to overcome the limitations of van der Waals contacts and achieve high-performance electronic devices.”

We have added the following new reference, Ref. [43] in the revised manuscript:

“43. Gao, L. *et al.* Atomic layer bonding contacts in two-dimensional semiconductors. *Science* **390**, 813–818 (2025).”

Comment 3: Temperatures in MD simulations: The selection and reporting of MD temperatures remain inconsistent. Temperatures listed in Table S1 should also appear clearly in the main text and relevant figure captions. Moreover, the rationale behind each temperature choice must be stated — currently, these choices appear arbitrary. This information is essential for readers to assess the physical relevance of the simulations.

Author reply: We apologize for the confusion regarding the temperature settings. We have now explicitly stated the simulation temperatures in the main text and all relevant figure captions. Furthermore, we have clarified the rationale for these choices in the Methods section, ensuring they systematically cover the **two-step vapor deposition** method reported in Ref. [23].

Specifically, **1100 K** was selected to align with **the sulfurization temperatures** (Step 2) of the experiment, where MoS₂ are synthesized at 800 °C (1073 K). **900 K** was chosen to correspond to the **pre-heating or baking stage** (experimentally >600 °C) and the heating ramp-up process prior to full sulfurization, allowing us to assess the stability of the metal layer as the system undergoes thermal treatment before reaching the peak reaction temperature.

In the revision, we revised the following sentences, on manuscript page 7:

“Subsequently, individual Mo atoms were simulated to be continuously sputtered onto the existing MoS₂ layer with a kinetic energy of 0.12 eV. To simulate the stability during the metal atom deposition and pre-heating stage, simulations were performed at 900 K.”

On manuscript page 19:

“For Mo atom deposition, we used a $4\sqrt{3} \times 7$ MoS₂ or WS₂ supercell at 900 K. This temperature was chosen to represent the thermal state during the pre-heating/baking stage and the ramp-up process prior to full sulfurization. The SMoMoS and SMMS sulfurized process simulations were performed using a $4\sqrt{3} \times 7$ supercell at 1100 K, aligning with the experimental sulfurization temperatures (approx. 1073 K).”

and added temperatures to all relevant figure captions (e.g., Fig. 2, Fig. 3, Fig. 4, etc.).

Comment 4: Reporting of reaction energies: Certain data remain ambiguously described or mislabeled. For instance, in Figure 2, the “sinking” of Mo is attributed to DFT calculations, although the associated energy change appears to originate from MD trajectories and was only later corroborated by DFT single-point calculations according to the SI(?). The authors must clarify which quantities stem from MD and which from DFT to prevent misinterpretation, alternatively, a description of the DFT calculations is required.

Author reply: We thank the reviewer for pointing out this ambiguity regarding the reporting of reaction energies. In the revised manuscript, we clarified that while the dynamic sinking process is illustrated by MLP-MD, the cited energy release of 1.45 eV was derived from DFT relaxations of the initial and final structures to ensure quantitative accuracy. Furthermore, all static reaction energy values reported throughout the manuscript are based on DFT calculations to ensure reliability (e.g., the energy differences for MoSMoS to SMOMoS, and MoSWS to SMOWS or SMMS).

In the revision, we revised the following sentences, on manuscript page 7:

“First, the adsorbed single Mo atom was unstable on the MoS₂ layer [Fig. 2(a)] and, at the typical growth temperature of 1100 K, quickly sank beneath the S atom layer within tens of picoseconds [Fig. 2(b)]. DFT calculations confirm that this embedding process releases 1.45 eV of energy.”

and

“DFT calculations reveal that the energy decreases significantly by 1.73 eV per Mo atom when transitioning from MoSMoS to SMOMoS, indicating a strong thermodynamic driving force towards the formation of the SMOMoS structure.”

On manuscript page 10:

“Conversely, the Mo atoms completely sink and embed under the top S layer of the WS₂ monolayer, forming the SMOWS intermediate structure [Fig. 3(h)], releasing 2.08 eV per Mo atom (based on DFT calculations).”

Comment 5: Interpretation of results: The authors’ additional discussion connecting this work to Ref. [23] is appreciated, but some confusion persists. It would be clearer to reference this work in the new paragraph as “In the above-mentioned study [23]” rather than “A recent study reported...” to guide readers properly, after all [23] is mentioned multiple times as a motivation for this study. More importantly, the authors’ findings still seem to contradict the experimental conclusions of Ref. [23], which reported successful wafer-scale synthesis of TMDs after direct dosing of Mo on WS₂. The authors could strengthen their argument by discussing potential reconciling factors. I would think a possible interpretation are sulfur-rich conditions that might have been present during experiments in [23] and thus stabilized against the intermediate SMeMeS

configuration, in accordance with the authors discussion in another context and findings around Figures S13–S16.

Author reply: We appreciate the reviewer’s insightful suggestion on reconciling our simulation results with the experimental findings in Ref. [23]. We have adopted the suggested phrasing “In the above-mentioned study [23]” in the revised manuscript to improve flow and clarity.

Regarding the experimental observations in Ref. [23], we note that the Mo film is deposited after the growth of the first WS₂ layer. Given the sulfur-rich environment of the preceding step, it is highly likely that residual sulfur remained in the growth chamber or on the substrate surface. This residual sulfur ensures sufficient sulfur availability, allowing Mo atoms to be partially sulfurized or pinned (forming Mo-S clusters) during deposition or the subsequent heating ramp-up, before they have a chance to sink. Our simulations (Fig. 4 and Fig. S13) explicitly confirm that once metal atoms bond with sulfur, their embedding behaviour is suppressed. Therefore, our work does not contradict the experimental results; rather, it provides an atomic-level mechanism: residual sulfur from the previous step is critical for suppressing the SMMS intermediate and preventing alloying. We have expanded the discussion in the revised manuscript to clarify this point.

In the revision, we revised the following sentence on manuscript page 11:

“In the above-mentioned study²³, a two-step vapor deposition process was reported to synthesize wafer-scale TMD vdWHs, where the key intermediate structure during growth is a Mo atomic monolayer on the WS₂ surface. However, our MLP-MD simulations reveal that the Mo atomic monolayer on the WS₂ surface is highly unstable, it can be easily transformed into the SMoWS structure by thermal annealing. This suggests that the experimentally observed clean interfaces must be governed by kinetics or environmental factors that prevent the formation of this bare metal intermediate.”

On manuscript page 14,

“Experimentally, the sulfur source and molybdenum source ratios used during the MOCVD growth of MoS₂ and WS₂ are significantly higher than 2:1, such as 70:1³⁸, 660:1³⁹, 6400:1⁴⁰, and 11111:1⁴¹. Similarly, regarding the successful synthesis in the above-mentioned study²³, we propose that sulfur-rich conditions likely played a decisive role. Since the Mo film is deposited after the synthesis of the first WS₂ layer, residual sulfur remaining in the growth chamber or on the surface from the preceding step is inevitable. This residual sulfur can react with deposited Mo atoms to form Mo-S clusters during the deposition or the initial heating phase, effectively suppressing the embedding of bare metal atoms and the formation of the SMMS intermediate, as predicted by our simulations. This mechanism explains how alloying is prevented in the experimental two-step process.”

Response to Reviewer 3

Comment: I co-reviewed this manuscript with one of the reviewers who provided the listed reports. This is part of the Nature Communications initiative to facilitate training in peer review and to provide appropriate recognition for Early Career Researchers who co-review manuscripts.

Author reply: We thank Reviewer 3 for co-reviewing the manuscript and for their time and contribution to the peer review process.

Response to Reviewer 4

Comment: Reviewer #4 (Remarks to the Author): The authors have significantly improved the manuscript and addressed the minor concerns I noted in my previous review. I'm happy to recommend the work for publication.

Reviewer #4 (Remarks on code availability): The code is now available on a public github repository and the changes made to the ocp package are clearer.

Author reply: We sincerely thank Reviewer 4 for the positive evaluation and for recommending our work for publication. We are glad that the revisions and the code availability have met the reviewer's expectations.

List of Changes

1. On Page 1 of the Supplementary Information, we have extensively revised the paragraphs describing the MLP training methodology to improve flow, terminology, and clarity. We have also added a new Figure S1(c) in the Supplementary Information (and a corresponding discussion on Page 5 of the Manuscript) to demonstrate the RMSE distribution as a function of structure formation energy, providing a comprehensive assessment of model convergence and quality. This is in response to Comment 1 from Reviewer 1.
2. On Page 20 of the Manuscript, we have added a statement confirming that key non-equilibrium processes (metal sinking and sulfurization) were verified in three independent MD simulations with different random seeds to ensure reproducibility. On Page 17, we have added a discussion and a new reference (Ref. [43]) regarding the "atomic layer bonding (ALB)" contact, which provides strong experimental corroboration for the stability and potential of our predicted SMMS intermediate. This is in response to Comment 2 from Reviewer 1.
3. On Page 7 and Page 19 of the Manuscript, we have clarified the rationale for simulation temperatures (1100 K and 900 K) in the Methods section, explicitly linking them to the sulfurization and pre-heating steps of the experimental method in Ref. [23]. We have also ensured that specific temperatures are clearly stated in all relevant figure captions throughout the manuscript. This is in response to Comment 3 from Reviewer 1.
4. On Pages 7 and 10 of the Manuscript, we have revised the text to explicitly distinguish between dynamic energy evolution (derived from MLP-MD) and quantitative reaction energy values (derived from static DFT calculations). This is in response to Comment 4 from Reviewer 1.
5. On Pages 11 and 14 of the Manuscript, we have expanded the discussion to propose an atomic-scale mechanism involving "residual sulfur" from the preceding growth step. This explains how sulfur-rich conditions in the experiment suppress metal embedding and alloying, thereby reconciling our simulation results with the experimental findings in Ref. [23]. We have also adopted the suggested phrasing "In the above-mentioned study [23]". This is in response to Comment 5 from Reviewer 1.